# CAR-neutrophil mediated delivery of tumor-microenvironment responsive nanodrugs for glioblastoma chemo-immunotherapy

Yun Chang[1,2], Xuechao Cai[3], Ramizah Syahirah [4], Yuxing Yao[5], Yang Xu[6], Gyuhyung Jin[1,2], Vijesh J. Bhute[7], Sandra Torregrosa-Allen[2], Bennett D. Elzey[2,8], You-Yeon Won [1,2], Qing Deng[2,4], Xiaojun Lance Lian [9,10,11] ✉, Xiaoguang Wang [6,12] ✉, Omolola Eniola-Adefeso [13] ✉ & Xiaoping Bao [1,2] ✉

Glioblastoma (GBM) is one of the most aggressive and lethal solid tumors in human. While efficacious therapeutics, such as emerging chimeric antigen receptor (CAR)-T cells and chemotherapeutics, have been developed to treat various cancers, their effectiveness in GBM treatment has been hindered largely by the blood-brain barrier and blood-brain-tumor barriers. Human neutrophils effectively cross physiological barriers and display effector immunity against pathogens but the short lifespan and resistance to genome editing of primary neutrophils have limited their broad application in immunotherapy. Here we genetically engineer human pluripotent stem cells with CRISPR/Cas9-mediated gene knock-in to express various anti-GBM CAR constructs with T-specific CD3ζ or neutrophil-specific γ-signaling domains. CAR-neutrophils with the best anti-tumor activity are produced to specifically and noninvasively deliver and release tumor microenvironment-responsive nanodrugs to target GBM without the need to induce additional inflammation at the tumor sites. This combinatory chemo-immunotherapy exhibits superior and specific anti-GBM activities, reduces off-target drug delivery and prolongs lifespan in female tumor-bearing mice. Together, this biomimetic CAR-neutrophil drug delivery system is a safe, potent and versatile platform for treating GBM and possibly other devastating diseases.

Glioblastoma (GBM) is characterized by a high mortality rate, short lifetime, and poor prognosis with a high tendency of recurrence[1,2]. Therapeutic efficacies of both surgery and chemo-drugs are primarily hindered by the fine brain structure and physiological blood-brain barrier (BBB) or blood-brain-tumor barrier (BBTB)[3–5]. Particularly, drug delivery to the central nervous system (CNS) for treating brain tumor is very challenging: <1% of administered nanoparticle dose is found to be delivered to a solid tumor based on 376 published datasets[6], and 0.8% delivered to brain cancer[7]. Due to their native capacity to migrate towards inflamed sites, traverse BBB/BBTB and infiltrate solid tumors, mouse neutrophil-mediated delivery of nanoparticulated chemo-

drugs has been investigated to enhance targeted drug delivery to the brain tumors for improved therapeutic efficacy[8–10]. However, an invasive surgical resection of the tumor or tumor microenvironment priming is needed to induce additional inflammation for neutrophil recruitment before neutrophil/chemotherapeutic administration, leading to the limited neutrophil recruitment in tumor sites beyond the inflamed surgical margin[11]. Furthermore, neutrophil-delivered chemotherapeutics were primarily enriched in the spleen, but not in the targeted brain of tumor-bearing mice. While necrosis was not observed in the major organs of experimental mice, there are still concerns regarding off-target tissue toxicity or even systemic toxicity

---

in patients[12]. Previous studies also focused on mouse neutrophils. The feasibility and safety of using human neutrophils in drug delivery remains elusive since neutrophils have a short lifespan and are prone to apoptosis ex vivo. In addition, massive neutrophil extraction from pre-surgical patients for drug loading may lead to neutropenia or other risks. Thus, a safe and effective human neutrophil-mediated biomimetic drug delivery system that utilizes the natural chemo-attractive GBM microenvironment is urgently needed.

Neutrophils' innate immunity and plasticity against various cancers[12–16], including GBM, were less explored than their application as cell carriers in drug delivery[8–10]. Circulating neutrophils in the blood home to the hypoxic tumor microenvironment (TME), where they become heterogenous tumor-associated neutrophils (TANs), an essential component of immunosuppressive TME that contributes to cancer progression and therapeutic resistance[12,17]. Similar to macrophages, anti-tumor N1 and pro-tumor N2 phenotypes of TANs were found within the hypoxic TME[18–21]. Various therapeutic strategies have been developed to directly target neutrophils with a focus on neutrophil depletion or inhibition[12,22], leading to several clinical trials (e.g., CCR5 inhibitor Maraviroc in NCT03274804). Thus, the direct application of untreated neutrophils as a nanocarrier may pose an additional risk to cancer patients in which drug-trafficking neutrophils may be reprogrammed to the immunosuppressive pro-tumor N2 phenotype within TME after homing to tumor sites[13,23]. In addition, the intrinsic anti-tumor activities of naïve neutrophils should be explored and boosted to achieve an optimized therapeutic efficacy when used as a drug carrier in combination with chemotherapeutics.

Chimeric antigen receptor (CAR) modification has significantly enhanced anti-tumor activities of immune T or natural killer (NK) cells[24–27]. However, their efficacy in solid tumors is still limited due in part to their relatively low trafficking and tumor penetration ability. The presence of physiological BBB and BBTB further impedes the efficacy of these emerging therapeutics against GBM in the brain. We speculated that the combination of CAR-engineering and highly motile neutrophils might sustain their anti-tumor N1 phenotype and yield excellent therapeutic efficacy in treating GBM. Primary neutrophils are short-lived and resistant to genome editing[28], limiting their application in CAR-directed immunotherapy. Human pluripotent stem cells (hPSCs), which are more accessible to gene editing and capable of differentiating into neutrophils massively, could provide an unlimited source of high-quality CAR-neutrophils for targeted immunotherapy under chemically-defined, xeno-free conditions[29]. Neutrophils also preferentially phagocytose microbial pathogens with rough or long surfaces, such as *S. aureus* and *E. coli*[30], which should be taken into account for nanoparticle design in neutrophil-mediated drug delivery. Indeed, Safari *et al.* recently reported the preferred phagocytosis of intravenously administered elongated particles, without complicated surface modification, by circulating neutrophils[30]. Such an easy and bioinspired design in drug-loaded nanoparticles may maximize drug-loading in neutrophils and allow therapeutic levels of drug delivery at targeted sites.

In this work, we design and screen four anti-GBM chlorotoxin (CLTX)-CAR constructs with T or neutrophil-specific signaling domains by knocking them into the *AAVS1* safe harbor locus of hPSCs via CRISPR/Cas9-mediated homologous recombination and identified an optimized CAR, composed of a 36-amino acid GBM-targeting CLTX peptide[27], a CD4 transmembrane domain and a CD3ζ intracellular domain, for neutrophil-mediated tumor-killing. The resulting stable CAR-expressing hPSCs are then differentiated into CAR-neutrophils, which sustain an anti-tumor N1 phenotype and exhibit enhanced anti-GBM activities under the hypoxic tumor microenvironment. A biodegradable mesoporous organic silica nanoparticle with a rough surface (R-SiO$_2$) is synthesized and employed to load hypoxia-activated pro-drug tirapazamine (TPZ) or clinical chemo-drug temozolomide (TMZ) and JNJ-64619187 (a potent PRMT5 inhibitor under clinical trial

NCT03573310) into hPSC-derived CAR-neutrophils, which are unharmed by the nanoparticulated cargo and retain the inherent biophysiological properties of naïve neutrophils. CAR-neutrophils loaded with drug-containing SiO$_2$ nanoparticles display superior anti-tumor activities against GBM, possibly due to a combination of CAR-enhanced direct cytolysis and chemotherapeutic-mediated tumor-killing via cellular uptake and glutathione (GSH)-induced degradation of nanoparticles within the targeted tumor cells. In an in situ GBM xenograft model, hPSC-derived CAR-neutrophils precisely and effectively deliver TPZ-loaded SiO$_2$ nanoparticles to the brain tumors without invasive surgical resection for amplified inflammation, significantly inhibit tumor growth, and prolong animal survival, representing a targeted and efficacious combinatory chemo-immunotherapy. Notably, Si content measurement suggests >20% of administered nanodrugs are delivered to brain tumor by CAR-neutrophils as compared to 1% by free nanodrugs. In summary, our biomimetic CAR-neutrophil drug delivery system is a safe, potent, and versatile platform for treating GBM and other devastating diseases.

## Results

### Screening neutrophil-specific CAR structures for enhanced anti-tumor activities

To engineer CAR-neutrophils for targeted drug delivery to brain tumor (Fig. 1a–b), we first designed and tested 4 different CAR structures optimized for anti-tumor activities of hPSC-neutrophils. All CAR structures shared the same extracellular granulocyte-macrophage colony-stimulating factor receptor (GM-CSFR) signal peptide (SP), glioblastoma-targeting domain CLTX[27], and IgG4 hinge[29] (Fig. 2a). CAR #1 is a first-generation T cell-specific CAR that uses the CD4 transmembrane (tm) domain and CD3ζ intracellular signaling domain. CAR #2, CAR #3, and CAR #4 differ from CAR #1 in using a transmembrane domain from neutrophil-specific CD32a (or FcγRIIA), a single-chain transmembrane receptor that is highly expressed in neutrophils (30,000 to 60,000 molecules/cell[31]) and critical for neutrophil activation[31–34]. CAR #3 and CAR #4 also include an Fc domain γ-chain of CD32a, which relies on a highly conserved immunoreceptor tyrosine-based activation motif (ITAM) to express and signal in neutrophils. Notably, CAR #3 contains a combo signaling domain by fusing CD32a-ITAM to the CD3ζ intracellular domain. Since primary neutrophils are short-lived and resistant to genome editing, we engineered human pluripotent stem cells (hPSCs) with these different CARs to achieve stable and universal immune receptor expression on differentiated neutrophils by knocking CAR constructs into the *AAVS1* safe harbor locus via CRISPR/Cas9-mediated homology-directed repair (Fig. 2b). After nucleofection, single cell-derived hPSC clones were isolated and screened with puromycin for about two weeks. Genotyping identified successfully targeted hPSCs with an average CAR knock-in efficiency of >90%, and the majority of targeted clones are heterozygous (Supplementary Fig. 1a–d). CAR expression on engineered hPSCs was further confirmed by RT-PCR and flow cytometry analysis of CLTX-IgG4 (Supplementary Fig. 1e–g). As expected, CAR-expressing hPSCs retained high expression levels of pluripotent markers, including OCT4, SSEA4 and SOX2 (Supplementary Fig. 1f).

To produce de novo CAR-neutrophils, CAR-expressing hPSCs were first differentiated into multipotent hematopoietic and then myeloid progenitors with stage-specific cytokine treatment[35] (Fig. 2c). Subsequent employment of G-CSF and retinoic acid agonist AM580 promoted robust neutrophil production[36]. Similar to their counterparts in peripheral blood (PB), hPSC-derived CLTX-CAR neutrophils presented typical neutrophil morphology and surface markers CD16, CD11b, MPO, CD15, CD66b, and CD18 (Supplementary Fig. 2). We next determined the effects of CAR expression on the anti-tumor cytotoxicity of hPSC-derived neutrophils by co-culturing them with glioblastoma (GBM) U87MG cells in vitro. As expected, hPSC-derived CLTX-CAR neutrophils presented improved tumor-killing

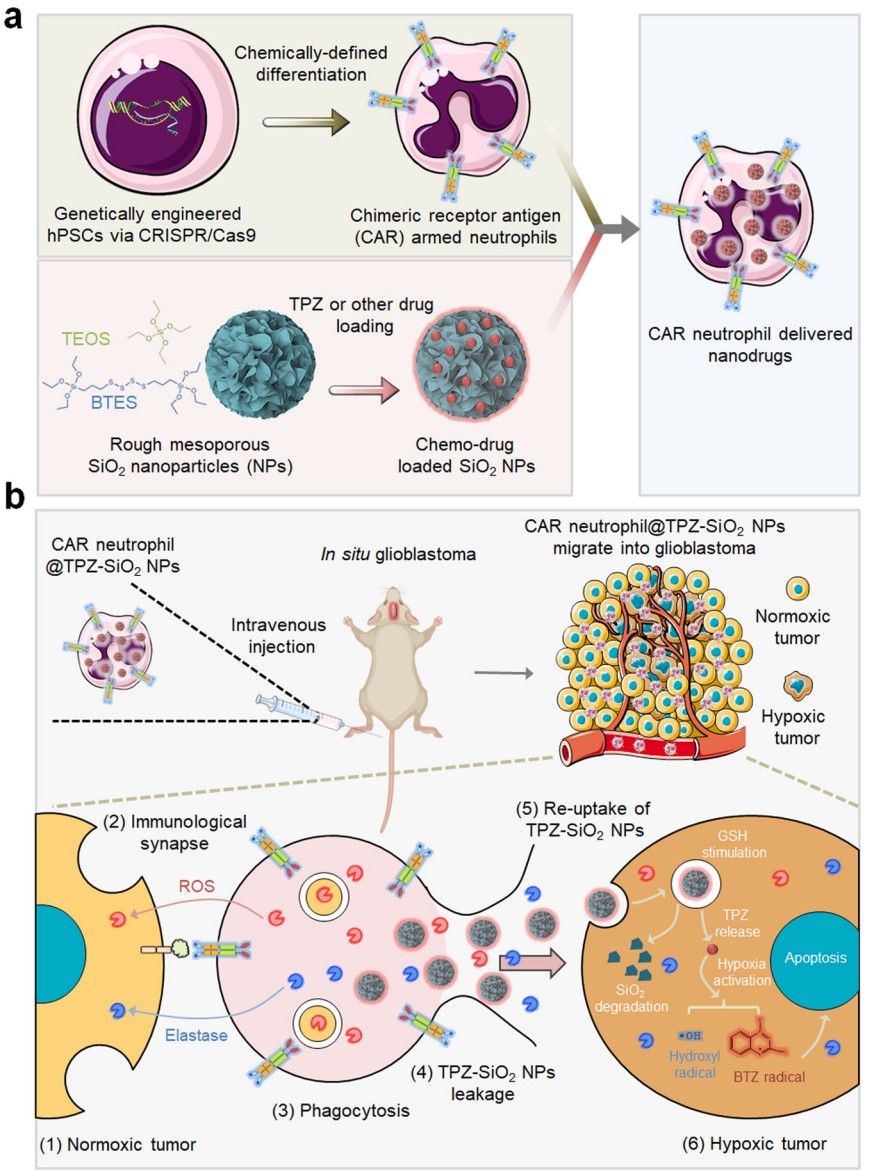

**Fig. 1 | Schematic of enhanced anti-glioblastoma efficacy using combinatory immunotherapy of CAR-neutrophils and tumor microenvironment responsive nanodrugs. a** Human pluripotent stem cells were engineered with CARs and differentiated into CAR-neutrophils that are loaded with rough silica nanoparticles (SiO₂ NPs) containing hypoxia-targeting tirapazamine (TPZ) or other drugs, as a dual immunochemotherapy. **b** Systemically administered CAR-neutrophil@R-SiO₂-TPZ NPs first attack external normoxic tumor cells by forming immunological synapses and kill tumor cells via phagocytosis. After apoptosis, CAR-neutrophils could then release R-SiO₂-TPZ NPs, which are uptaken by tumor cells. Afterwards, nano-prodrugs respond to the hypoxic tumor microenvironment and effectively kill tumor cells. TEOS tetraethyl orthosilicate, BTES bis[3-(triethoxysilyl) propyl] tetrasulfide, TPZ tirapazamine, BTZ benzotriazinyl.

ability as compared to PB neutrophils (Fig. 2d), consistent with previous observations in CLTX CAR-T cells[27]. Among these different CARs, CAR #1 mediated superior tumor-killing activities in hPSC-neutrophils. Notably, γ-chain-based CAR #4 is least effective in triggering neutrophil-mediated tumor-killing, which may be due to the lower copy of ITAM in γ than ζ-subunit and lower expression of γ-bearing CARs on the cell surface[28]. Neutrophils release cytotoxic reactive oxygen species (ROS) and tumor necrosis factor-α (TNF-α) to kill target cells. The production of ROS and TNF-α (Fig. 2e, f) from different neutrophils coincided well with their increased cytolysis. As expected, the production of ROS and TNF-α from different neutrophils after coculturing with normal SVG p12 glial cells remained as low as the negative control group (Supplementary Fig. 3a, b). In addition, enhanced anti-tumor cytotoxicity of CAR-neutrophils was only observed in co-incubation with GBM cells, including U87MG, primary adult GBM43, and pediatric SJ-GBM2 cells (Supplementary Fig. 3c),

demonstrating the high specificity of our CLTX-CAR. Notably, CAR-neutrophils exhibited high biocompatibility with normal SVG p12 glial cells, hPSCs, and hPSC-derived cells (Supplementary Fig. 3d), consistent with a previous observation that inactivated primary neutrophils do not kill healthy cells[16]. Collectively, hPSC-derived CAR-neutrophils, particularly CD3ζ-bearing CAR-neutrophils, presented enhanced anti-tumor cytotoxicity and produced more ROS and TNF-α in vitro, highlighting their potential in targeted immunotherapy.

## CAR-neutrophils sustained superior anti-tumor activities under immunosuppressive tumor microenvironments

Similar to macrophages, anti-tumor N1 and pro-tumor N2 phenotypes of tumor-associated neutrophils were found within the immunosuppressive tumor microenvironment[17]. Pro-tumor N2 neutrophils play critical roles in tumor angiogenesis, metastasis and immunosuppression, but therapeutic targeting of this cell type has been challenging.

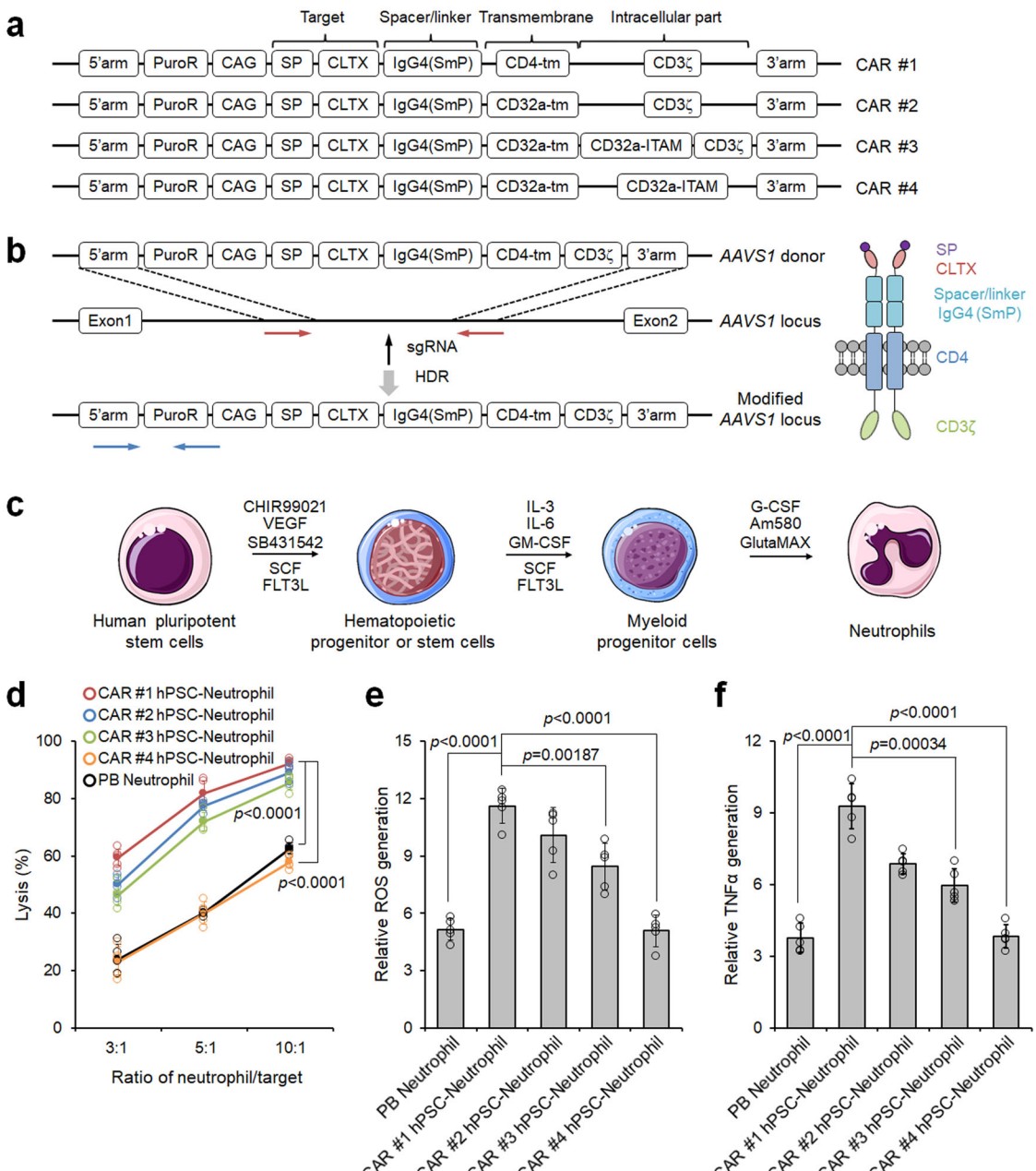

**Fig. 2 | Screening neutrophil-specific chimeric antigen receptor (CAR) structures with enhanced neutrophil-mediated anti-tumor activities. a** Schematic of various CAR structures. **b** Schematic of CAR #1 construct and targeted knock-in strategy at the *AAVS1* safe harbor locus of human pluripotent stem cells (hPSCs). Vertical arrow indicates the *AAVS1* targeting sgRNA. Red and blue horizontal arrows indicate primers for assaying targeting efficiency and homozygosity, respectively. HDR: homologous recombination repair. **c** Schematic of optimized neutrophil differentiation from hPSCs under chemically-defined conditions. **d** Cytotoxicity assays against U87MG glioblastoma cells were performed at different ratios of neutrophil-to-tumor target using indicated neutrophils. Data are represented as mean ± SD of five independent biological replicates, two-tailed Student's *t* test. Reactive oxygen species (ROS) generation (**e**) and ELISA analysis of TNFα release (**f**) from different neutrophils after coculturing with U87MG cells were determined. *n* = 5 biologically independent samples. The data are represented as mean ± SD, two-tailed Student's *t* test. Source data are provided as a Source Data file.

Rather than systemic depletion strategy[22], here we evaluated the potential of CAR-engineering in sustaining the anti-tumor phenotype of neutrophils. CAR hPSC-derived and PB neutrophils were treated with hypoxia (3% $O_2$) and TGFβ, which contribute to the immunosuppression of tumor microenvironment[37,38], to assess their sustained tumor-killing activity. While PB neutrophils presented significantly decreased cytolysis against GBM cells under immunosuppressive conditions, CAR-neutrophils sustained high tumor-killing activities (Supplementary Fig. 4a). Similar observations were also made in the TNFα release and ROS generation (Supplementary Fig. 4b, c) from PB

or CAR-neutrophils under immunosuppressive and normal conditions. To further confirm neutrophil phenotype under hypoxic and TGFβ conditions, we measured the expression of N1-specific iNOS and N2-specific arginase on the isolated neutrophils by flow cytometry (Supplementary Fig. 4d–f). Compared with normoxia, immunosuppressive hypoxia and TGFβ significantly decreased expression levels of iNOS and increased levels of arginase in PB neutrophils, whereas CAR-neutrophils retained high expression levels of iNOS. Previous studies indicate activation of Syk-Erk signaling pathway leads to the ROS production[39–42]. Therefore, we detected and compared Syk-Erk

activation in unmodified neutrophils and CAR-neutrophils, and our results suggested a significantly higher activation of Syk-Erk pathway in CAR-neutrophils under hypoxia (Supplementary Fig. 5a–d), which may sustain the unchanged ROS production of CAR-neutrophils under hypoxia. Taken together, CAR-neutrophils sustained an anti-tumor phenotype and maintained high anti-tumor activities under tumor microenvironment mimicking conditions in vitro, highlighting their potential in targeted immunotherapy.

## Preparation and characterization of hPSC CAR-neutrophils loaded with tirapazamine (TPZ)-containing SiO₂ nanoparticles

PB neutrophils have been used as cellular carriers to deliver imaging and therapeutic drugs into brain tumors[8–10], though targeted neutrophil infiltration requires surgery- or light-induced inflammation and off-target drug delivery may be a concern[11]. To further improve the anti-tumor activities of CAR-neutrophils, we prepared silica nanoparticles (SiO₂-NP) with a rough or smooth surface to load chemotherapeutic or radiation drugs into neutrophils. Transmission electron microscope (TEM) images demonstrated that both SiO₂ nanoparticles were well-dispersed and exhibited spherical morphology with a uniform size (Fig. 3a, Supplementary Fig. 6a). Composition distribution analysis via scanning TEM (STEM) with energy-dispersed X-ray spectroscopy (EDS) showed that the sulfur (S) element was evenly distributed within the whole rough SiO₂ nanoparticles (R-SiO₂) (Fig. 3b). Using nitrogen ($N_2$) adsorption-desorption isotherms and corresponding pore size distribution analysis, pore sizes of R- and S-SiO₂ NPs were measured as 25 nm and 35 nm (Fig. 3c, Supplementary Fig. 6b), respectively. Given the high surface area and large pore size, therapeutic drugs could be effectively loaded into both R- and S- SiO₂ NPs, as exemplified by the hypoxia-responsive pro-drug tirapazamine (TPZ) (Fig. 3d, Supplementary Fig. 6c). After TPZ loading, significant changes were not observed in the dispersity, morphology, and size of R-SiO₂-TPZ using TEM and dynamic light scattering analysis (Supplementary Fig. 6d, e). The tetra-sulfide bonds incorporated into the R-SiO₂ NPs are sensitive to reductive environments and can be rapidly degraded by the large amount of glutathione (GSH) present within the tumor cells[43]. We next determined GSH responsive degradability of R-SiO₂−TPZ NPs in the presence of 10 mM, 1 mM, and 10 µM GSH, which were the same as the intracellular conditions of cancer cells, normal cells, and extracellular environments[43], respectively. Upon 10 mM GSH treatment, the initial spherical structure of R-SiO₂−TPZ NPs was severely destructed after 24 hr (Supplementary Fig. 6f, g). The nanoparticles were completely disintegrated into small debris after 48 h, resulting in the TPZ release in a GSH-responsive manner (Fig. 3e). The debris of R-SiO2 NPs did not cause any significant cytotoxicity to the tested cells in vitro (Supplementary Fig. 6h), indicating the relative safety of R-SiO2 NPs.

We next evaluated the feasibility of using SiO₂−TPZ NPs to load therapeutic drugs into CAR-neutrophils as a combinatory chemoimmunotherapy to achieve boosted therapeutic efficacy. After centrifugation, we measured the cellular uptake of SiO₂−TPZ NPs by neutrophils using fluorescence microscope and flow cytometry analysis (Fig. 3f, g), and detected a more significant cellular uptake of R-SiO₂−TPZ NPs than S-SiO₂−TPZ NPs by neutrophils. Cellular Si content in neutrophils was measured as 11.3 and 19.1 ng Si/µg protein for smooth and rough SiO₂ NPs@TPZ (Fig. 3h), respectively, by inductively coupled plasma mass spectrometry (ICP-MS). Given their high loading capacity in neutrophils, R-SiO₂−TPZ NPs were employed for subsequent experiments.

We then sought to test the physiological functions of CAR-neutrophils after loading R-SiO₂−TPZ NPs. No changes were observed in cell viability (Fig. 3i, Supplementary Fig. 6i), transwell migration ability (Fig. 3j), chemotaxis and corresponding velocity (Fig. 3k, l) of CAR-neutrophils before or after loading R-SiO₂−TPZ NPs, demonstrating their high biocompatibility. Time-dependent nano-drug

loading analysis was also performed and the maximum loading content was reached at 1 h after cell-NP incubation (Supplementary Fig. 7a). More than 95% CAR-neutrophils were successfully loaded with R-SiO₂−TPZ NPs (Supplementary Fig. 7b). The expression level of CD11b, a neutrophil surface protein that mediates adhesion and migration function upon inflammatory molecule stimulation, was not changed on CAR-neutrophils with or without R-SiO₂-TPZ loading (Supplementary Fig. 7c, d). Superoxide or reactive oxygen species (ROS) are released from active neutrophils to kill microbes and tumor cells[44]. As expected, ROS generation by CAR-neutrophils was significantly increased after N-Formylmethionine-leucyl-phenylalanine (fMLP) treatment, and significant differences were not observed in ROS production by CAR-neutrophils before and after loading R-SiO₂-TPZ (Fig. 3m). Taken together, our data demonstrated that R-SiO₂-TPZ loaded CAR-neutrophils maintained the physiological activities of wild-type neutrophils and could actively migrate towards inflammatory stimuli, highlighting their potential in targeted cancer chemoimmunotherapy.

## CAR-neutrophils loaded with R-SiO₂-TPZ nanoparticles effectively kill glioblastoma cells

We next evaluated the effect of R-SiO2-TPZ on the tumor-killing ability of CAR-neutrophils. Intimate effector-target interaction was a prerequisite for neutrophil-mediated cytolysis. As expected, CAR-neutrophils@R-SiO₂-TPZ formed immune synapses with tumor cells within 2 h and exhibited similar effector-target interaction numbers as drug-free CAR-neutrophils (Fig. 4a, Supplementary Fig. 8). Notably, no observable interactions were found between CAR-neutrophils@R-SiO₂-TPZ and noncancerous somatic cells (Supplementary Fig. 8), highlighting the specificity of CLTX-CAR against brain tumors. Furthermore, R-SiO₂−TPZ NPs were released from neutrophils into the culture medium (Supplementary Fig. 9a, b) 12 h after co-culture and entered the remaining tumor cells (Fig. 4a). Twenty-four hours after co-incubation of SiO₂−TPZ NP-loaded CAR-neutrophils with tumor cells, up to 95% of tumor cells contained R-SiO₂−TPZ NPs (Fig. 4a, Supplementary Fig. 9c), indicating a successful transport cascade involving carrier neutrophils that exert their effector cell function and undergo apoptosis, thereby passively releasing R-SiO₂−TPZ NPs to the target tumor cells[45]. We also validated hypoxic responsive function and cytotoxicity of pro-drug TPZ within tumor cells by electron paramagnetic resonance (EPR) spectroscopy analysis of radicals generation from TPZ (Supplementary Fig. 9d) and flow cytometry analysis of TO-PRO-3 on tumor cells (Supplementary Fig. 9e) under hypoxia and normoxia. To determine the cytolysis of R-SiO₂−TPZ NP-loaded CAR-neutrophils, we implemented an in vitro normoxia-hypoxia tumor rechallenging model (Fig. 4b). Twenty-four hours after normoxic coculture, CAR-neutrophils loaded with R-SiO₂-TPZ NPs or not exhibited similar anti-tumor cytotoxicity (Fig. 4c), and both were higher than that of PB-neutrophil loaded with R-SiO₂-TPZ NPs or not and R-SiO₂-TPZ NPs alone. The enhanced cytotoxicity is mainly due to the increased tumor-targeting ability of neutrophils after CAR engineering. After an additional 12 and 24-h hypoxic coculture with tumor cells, R-SiO₂-TPZ NP-loaded CAR-neutrophils displayed superior anti-tumor ability compared to other groups (Fig. 4d, e). In addition, CAR-neutrophils loaded with R-SiO₂-TPZ NPs exhibited excellent cytolysis against re-seeded fresh tumor cells (Fig. 4f), indicating the anti-tumor ability of released R-SiO₂-TPZ nanoparticles after neutrophil apoptosis.

We then performed RNA sequencing (RNA-seq) analysis on tumor cells to elucidate the potential molecular mechanism underlying enhanced anti-tumor cytolysis of neutrophils by CAR expression and R-SiO₂-TPZ NPs. Gene expression analysis demonstrated that compared to control and R-SiO₂-TPZ NPs, CAR-neutrophils loaded with or without R-SiO₂-TPZ NPs significantly decreased the expression of cytoplasm and membrane genes in tumor cells (Supplementary Fig. 10a, Fig. 4g), further supporting their phagocytosis of tumor cells

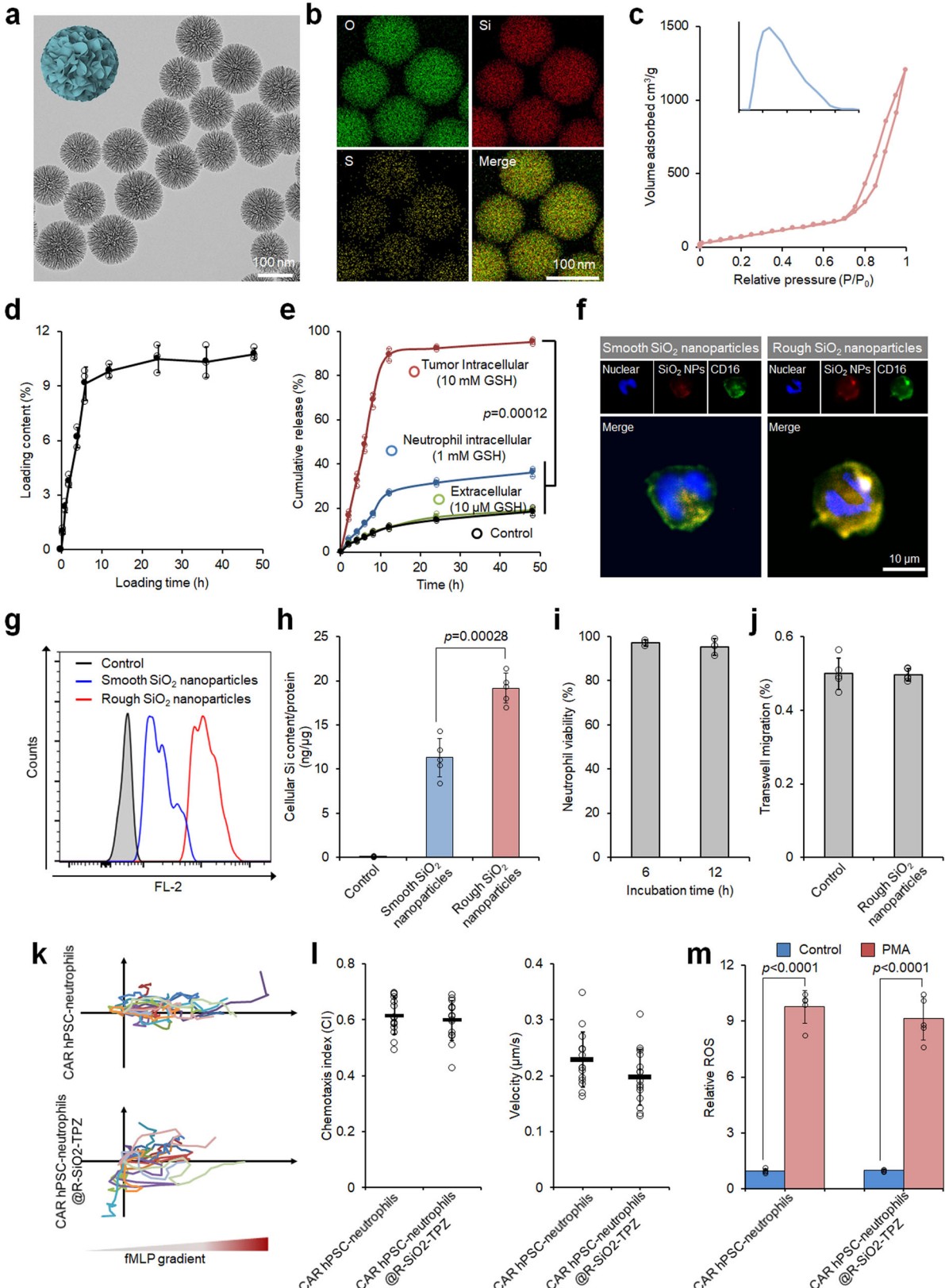

upon coculture. While all experimental groups increased cellular oxidative stress in tumor cells, R-SiO₂-TPZ-loaded CAR-neutrophils outperformed other groups in triggering oxidative stress signaling. In addition, R-SiO₂-TPZ-loaded CAR-neutrophils significantly promoted apoptosis and decreased proliferation in tumor cells. To further understand the enhanced anti-tumor activities of R-SiO₂-TPZ-loaded

CAR-neutrophils, we applied a phagocytosis inhibitor cytochalasin D and a reactive oxygen species (ROS) scavenger N-acetyl-cysteine (NAC) and a ROS inhibitor GSK2795039 to the tumor-neutrophil coculture. Cytolysis of tumor cells by CAR-neutrophils was significantly reduced by 5 μM cytochalasin D, 5 mM NAC and 100 nM GSK2795039 (Supplementary Fig. 10b, c), indicating the prominent role of phagocytosis

**Fig. 3 | Preparation and characterization of hPSC CAR-neutrophils loaded with tirapazamine (TPZ)-containing SiO$_2$ nanoparticles. a–e** Transmission electron microscope (TEM) (**a**) and energy dispersive spectroscopy (EDS) elemental mapping images (**b**) of rough SiO$_2$ nanoparticles are shown. **c** Nitrogen adsorption-desorption isotherm of rough SiO$_2$ nanoparticles along with Barrett-Joyner-Halenda (BJH) pore size distribution plot is shown. Biological triplicates were performed independently. TPZ loading content in SiO$_2$ nanoparticles (**d**) and glutathione (GSH)-responsive TPZ release (**e**) were measured at the indicated time. $n = 3$ biologically independent samples. One-way analysis of variance (ANOVA) for (**e**). Fluorescence images (**f**) and flow cytometry analysis (**g**) of neutrophils loaded with smooth and rough SiO$_2$-TPZ. Biological triplicates were performed independently. **h** Cellular SiO$_2$ content in hPSC-derived CAR-neutrophils was measured. $n = 5$ biologically independent samples, two-tailed Student's $t$ test. Cellular viability (**i**), $n = 3$ biologically independent samples, transmigration (**j**), $n = 5$ biologically independent samples, chemoattraction abilities (**k**, **l**), $n = 20$ biologically independent samples, and ROS generation ability (**m**) of hPSC-derived CAR-neutrophils loaded with or without rough SiO$_2$-TPZ were shown, $n = 5$ biologically independent samples, two-tailed Student's $t$ test. PMA: phorbol myristate acetate. All data in this figure are represented as mean ± SD. Source data are provided as a Source Data file.

and ROS in CAR neutrophil-mediated tumor-cell killing. The remaining 40%-50% tumor cell lysis in the presence of neutrophils and NAC or GSK2795039 indicates the involvement of ROS independent mechanism in neutrophil-mediated tumor-killing that is worth further investigation.

## Functional evaluation of CAR-neutrophils loaded with nanodrugs using biomimetic glioblastoma models in vitro

To further assess the activities of R-SiO$_2$-TPZ NP-loaded CAR-neutrophils, we implemented a transwell-based blood-brain barrier (BBB) tumor model using human cerebral microvascular endothelial cells (Fig. 5a, Supplementary Fig. 11a). As expected, R-SiO$_2$-TPZ NP-loaded CAR-neutrophils exhibited excellent transmigration ability across in vitro BBB model (Fig. 5b), effectively killing targeted tumor cells after transmigration under both normoxic and hypoxic conditions (Fig. 5c, d), and releasing more inflammatory cytokines (Fig. 5e) that may attract other effector cells to kill the tumor cells. In addition, CAR-neutrophils did not significantly affect viability of endothelial cells after transmigration (Supplementary Fig. 11b). R-SiO$_2$-TPZ NP-loaded CAR-neutrophils retained excellent transmigration ability during the second transmigration experiment (Fig. 5f) and superior anti-tumor ability compared with other groups (Fig. 5g). A three-dimensional (3D) tumor spheroid model was then employed to evaluate the tumor-penetration capacity of R-SiO$_2$-TPZ NP-loaded CAR-neutrophils (Fig. 5h). CAR-neutrophils gradually migrated towards the center of the tumor spheroid and uniformly distributed in the spheroid after 8 h of incubation (Fig. 5i). A high degree of co-localization between CAR-neutrophils and R-SiO$_2$-TPZ NPs was observed (Supplementary Fig. 12a–c), demonstrating that R-SiO$_2$-TPZ NPs were encapsulated stably in the CAR-neutrophils during tumor infiltration before their cytolysis. Without neutrophil-mediated delivery, R-SiO$_2$-TPZ NPs were only found on the outside layer of tumor spheroids. Compared to R-SiO$_2$-TPZ NPs and CAR-neutrophils, R-SiO$_2$-TPZ NP-loaded CAR-neutrophils exhibited superior anti-tumor cytolysis in the 3D tumor model (Fig. 5j). CAR-neutrophils@R-SiO$_2$ NPs can also be employed to deliver other drugs, including clinical temozolomide (TMZ) and JNJ-64619187, into 3D tumor models and efficiently kill GBM cells (Supplementary Fig. 12d–f). Taken together, the combinatory CAR-neutrophils and nanodrugs displayed excellent anti-tumor activities in biomimetic tumor microenvironment mimicking conditions in vitro, highlighting the therapeutic potential of combinatory neutrophil-based chemoimmunotherapy.

## In vivo distribution of CAR neutrophil-delivered R-SiO$_2$-TPZ nanoparticles

In addition to improving the direct tumor-killing ability, we hypothesize that CAR engineering of hPSC-neutrophils will significantly enhance their targeted delivery of therapeutic drugs without additional surgery- or light-induced inflammation[11]. To test this hypothesis, we employed a mouse xenograft model of glioblastoma and an in vivo imaging system to determine the trafficking and biodistribution of R-SiO$_2$-TPZ NP-loaded CAR-neutrophils. We fluorescently labeled SiO$_2$ NPs with a near-infrared dye Cyanine 5 (Cy5) and then performed fluorescence imaging 3 h and 24 h after systemic administration

(Fig. 6a). Three hours after intravenous injection, R-SiO$_2$-TPZ NPs traveled to the whole body of tumor-bearing mice and emitted strong fluorescence with or without neutrophil-mediated delivery (Fig. 6b). CAR-neutrophil-delivered R-SiO$_2$-TPZ NPs accumulated in the brain tumor site within 24 h, whereas free R-SiO$_2$-TPZ NPs were still evenly distributed across the whole body (Fig. 6b). To further quantify the biodistribution of R-SiO$_2$-TPZ NPs in various organs, inductively coupled plasma-optical emission spectrometry (ICP-OES) analysis of Si content was performed on the harvested organs 24 h post-injection. CAR neutrophil-delivered R-SiO$_2$-TPZ NPs were significantly enriched in the mouse brain (Fig. 6c), although a low-level delivery to the liver and spleen was observed. Si content measurement also demonstrated that >20% of administered nanodrugs were delivered to brain tumor by CAR-neutrophils as compared to 1% by free nanodrugs that is consistent with previous reports[6]. Targeted delivery of R-SiO$_2$-TPZ NPs to the host brain across BBB by CAR-neutrophils was also confirmed by histology analysis (Fig. 6d). On the contrary, R-SiO$_2$-TPZ NPs alone mainly accumulated in the liver and spleen. Collectively, our data demonstrated enhanced targeted delivery of R-SiO$_2$-TPZ NPs by CAR-neutrophils without the need to induce additional inflammation at the tumor site, highlighting the feasibility and safety of neutrophil-based chemoimmunotherapy in cancer treatment.

## The combinatory chemoimmunotherapy of CAR-neutrophils and R-SiO$_2$-TPZ nanoparticles exhibited excellent anti-glioblastoma activities in vivo

To determine the therapeutic efficacy of R-SiO$_2$-TPZ NP-loaded CAR-neutrophils, in situ xenograft model of glioblastoma was established in the NOD.Cg-$RAG^{tm1Mom}IL2rg^{tm1Wjl}$/SzJ (NRG) mice using luciferase-expressing U87MG cells. Tumor-bearing mice were intravenously administrated $5 \times 10^6$ neutrophils weekly (Fig. 7a), and tumor burden in the hosts was measured and quantified (Fig. 7b, c). Compared to PBS or PB-neutrophil-treated mice, treatment with CAR-neutrophils and CAR-neutrophil@R-SiO$_2$–TPZ NPs effectively slowed tumor growth. CAR-neutrophils@R-SiO$_2$–TPZ NPs displayed much higher anti-tumor cytotoxicity than any other experimental groups. On the contrary, PB-neutrophils significantly promoted tumor growth in the brain, resulting in the death of tumor-bearing mice as early as day 23 (Fig. 7d), suggesting that unengineered neutrophils may pose additional risks. We next measured human cytokine release in the plasma of different experimental mouse groups (Fig. 7e). All non-PBS experimental groups produced detectable TNFα and IL-6 in the plasma from day 5 to day 26, suggesting the activation of human neutrophils upon tumor stimulation. Consistent with observed higher tumor growth rate, unmodified neutrophils gradually released more IL-6 and TNFα, which may lead to cytokine release syndrome in patients and require more in-depth safety studies with IL-6 blockers[46,47]. Notably, CAR-neutrophils@R-SiO$_2$–TPZ NPs displayed decreased cytokine production ability at later time points (day 19 and day 26), suggesting a potentially low risk of cytokine release syndrome in patients treated with CAR-neutrophil-based chemoimmunotherapy.

The biocompatibility of combinatory CAR-neutrophils and R-SiO$_2$–TPZ NPs was evaluated through weekly measurement of body weights and monitoring of pathological changes in major organs of

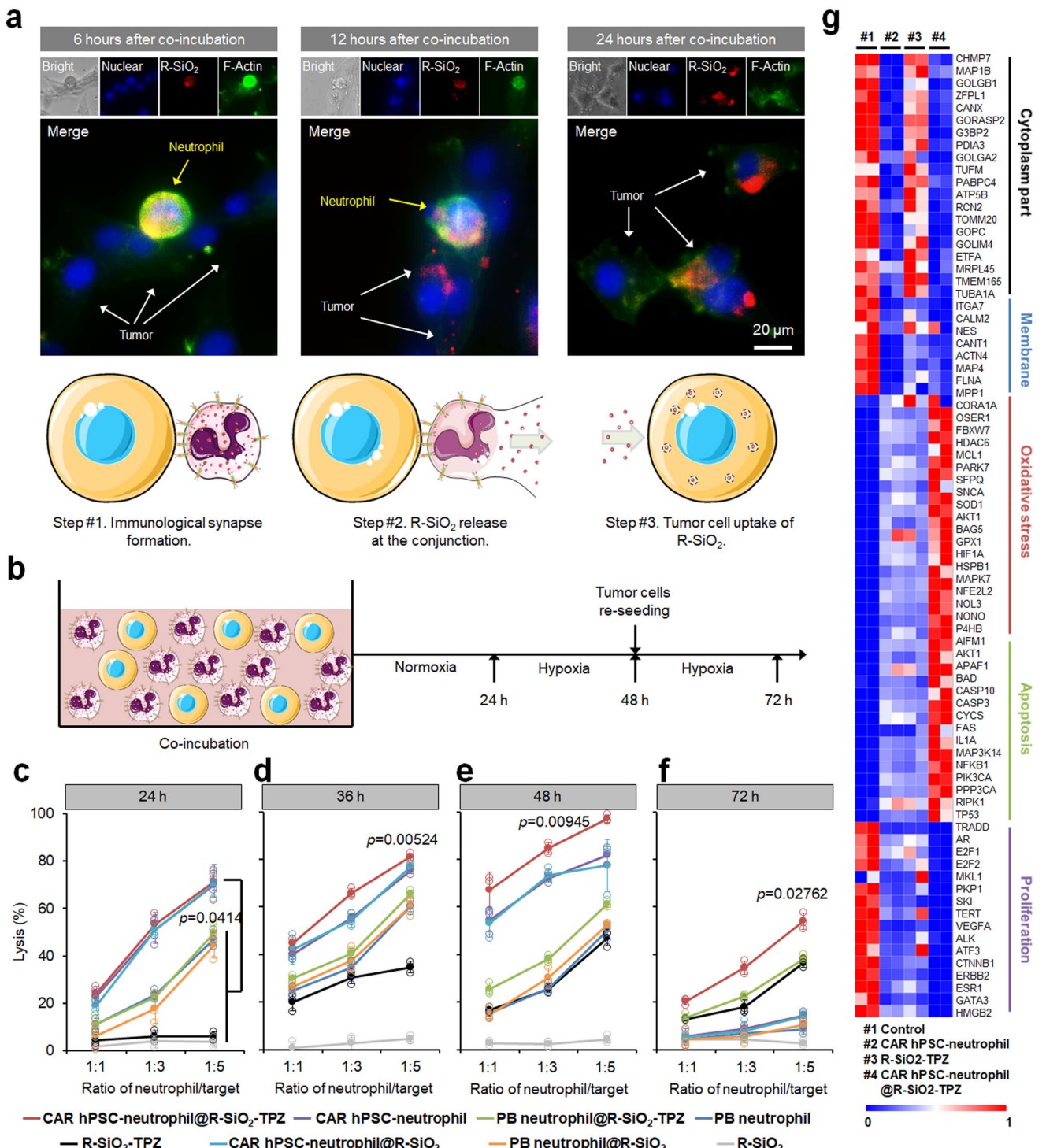

**Fig. 4 | CAR-neutrophils loaded with R-SiO₂-TPZ nanoparticles effectively kill glioblastoma cells. a** Representative images of immunological synapses indicated by polarized F-actin accumulation at the interface between CAR-neutrophils and tumor cells at 6, 12 and 24 h were shown. R-SiO₂-TPZ nanoparticles released from CAR-neutrophils upon tumor cell phagocytosis were up-taken by tumor cells. Triplicates were performed independently. **b** Schematic of neutrophil-mediated anti-tumor cytotoxicity assay. Cytotoxicity against U87MG glioblastoma cells were performed at different ratios of neutrophil-to-tumor target using indicated neutrophils at 24 h (**c**), 36 h (**d**), 48 h (**e**), and 72 h (**f**). $n = 3$ biologically independent samples. Data are represented as mean ± SD, one-way analysis of variance (ANOVA). **g** Bulk RNA sequencing analysis was performed on U87MG cells under various conditions. Heatmap shows expression levels of selected cytoplasm, membrane, oxidative stress, apoptosis, and proliferation-related genes in the indicated glioblastoma cells. $n = 2$ biologically independent samples. Source data are provided as a Source Data file.

mice. No difference was observed in body weights between CAR-neutrophils@R-SiO₂−TPZ NP-treated mice and any other experimental groups (Fig. 7f), indicating minimal systemic toxicity and excellent biocompatibility of CAR-neutrophils@ R-SiO₂−TPZ NPs within 28 days

of treatment. Histological analysis on major organs sliced from mice on day 30 showed that CAR-neutrophils@R-SiO₂−TPZ NP-treated mice did not cause noticeable abnormality or organ damage in the heart, liver, spleen, lung, and kidney (Supplementary Fig. 13), further

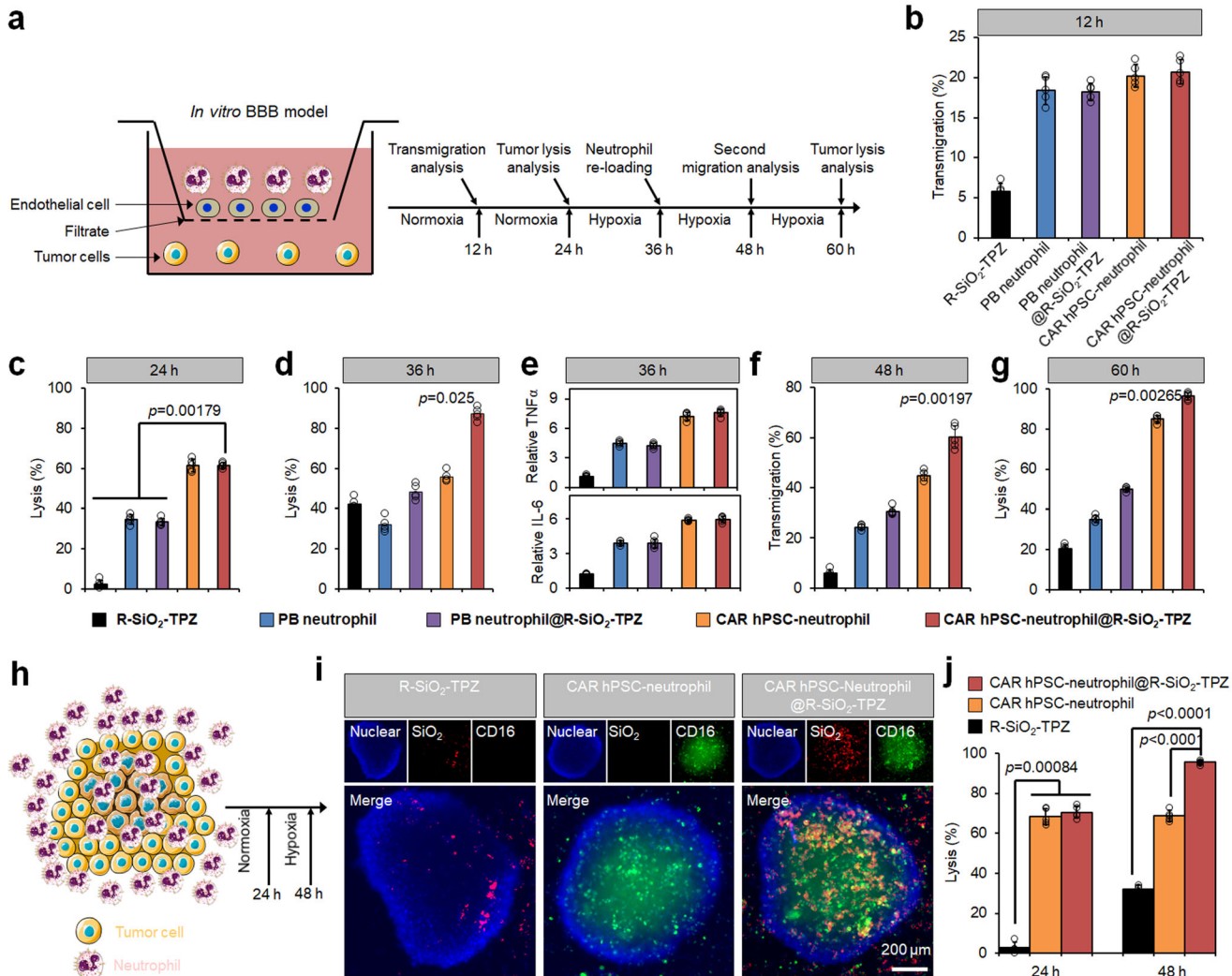

**Fig. 5 | Functional evaluation of CAR-neutrophils loaded with R-SiO$_2$-TPZ nanoparticles using biomimetic glioblastoma (GBM) models in vitro.**
**a** Schematic of our in vitro tumor model of GBM with blood-brain-barrier (BBB), which is composed of endothelial cells on the cell insert membrane and tumor cells in the bottom of the same transwell. **b** Transwell migration analysis of neutrophils at 12 h is shown. Anti-GBM cytotoxicity of indicated neutrophils at 24 h (**c**) and 36 h (**d**) was measured and quantified. **e** ELISA analysis of IL-6 and TNFα released from indicated neutrophils at 36 h was performed. **f** Second migration of different neutrophils at 48 h is shown. **g** Anti-GBM cytotoxicity of indicated neutrophils at 60 h was measured and quantified. **h–j** Schematic of neutrophil-infiltrated three-dimensional (3D) tumor model in vitro was shown in (**h**). **i** Representative fluorescent images of infiltrated neutrophils in the 3D tumor models were shown. DAPI was used to stain the cell nuclear and CD45 was used to stain neutrophils. Scale bars, 200 μm. Biological triplicates were performed independently. **j** The corresponding tumor-killing ability of indicated neutrophils was measured and quantified using cytotoxicity kit. Data are represented as mean ± SD of five independent biological replicates, one-way analysis of variance (ANOVA). Source data are provided as a Source Data file.

confirming the safety of combinatory CAR-neutrophils and R-SiO$_2$–TPZ NPs.

While CAR-neutrophil@R-SiO$_2$-TPZ NPs significantly slowed down tumor growth in xenograft mice, the difference of animal survival in experimental groups of CAR-neutrophils, SiO$_2$–TPZ NPs and CAR-neutrophil@R-SiO$_2$–TPZ NPs is insignificant ($p > 0.05$), which is possibly due to the death of short-lived neutrophils during cell preparation and injection. We next focused on these three groups and determined if reduced cell preparation time and increased dosages of CAR-neutrophils and nanodrugs would make any difference in animal survival (Fig. 7g). When systemically administered 6 times, CAR-neutrophil@R-SiO$_2$–TPZ NPs outperformed the other two groups in extending lifespan of tumor-bearing mice (Fig. 7h), whereas the difference of animal survival in groups of CAR-neutrophils and SiO$_2$–TPZ NPs remained insignificant. While a similar survival curve of the R-SiO2-TPZ group was observed between these two independent animal studies, reduced time in cell isolation and preparation for injection from a

total of ~4 hrs to 1 hr during the first 4 neutrophil doses led to improved animal survival in CAR-neutrophil groups before day 32. Collectively, our data demonstrated the importance of neutrophil preparation and dosage optimization in future clinical application of neutrophil therapeutics.

## Discussion

Mouse neutrophils have been demonstrated as a powerful carrier to efficiently deliver nanodrugs to the inflamed post-operative brain tumors[8,9]. Still, the feasibility and safety of using human neutrophils in drug delivery remain elusive. The large amount of mouse neutrophils (10 times higher than total number of circulating neutrophils in mice[11]) used in these studies to achieve therapeutic benefit may further hinder their clinical translation since the extraction of large numbers of neutrophils from cancer patients may lead to neutropenia and pose other risks. To address these challenges, we harnessed the power of self-renewing hPSCs in obtaining unlimited de novo human

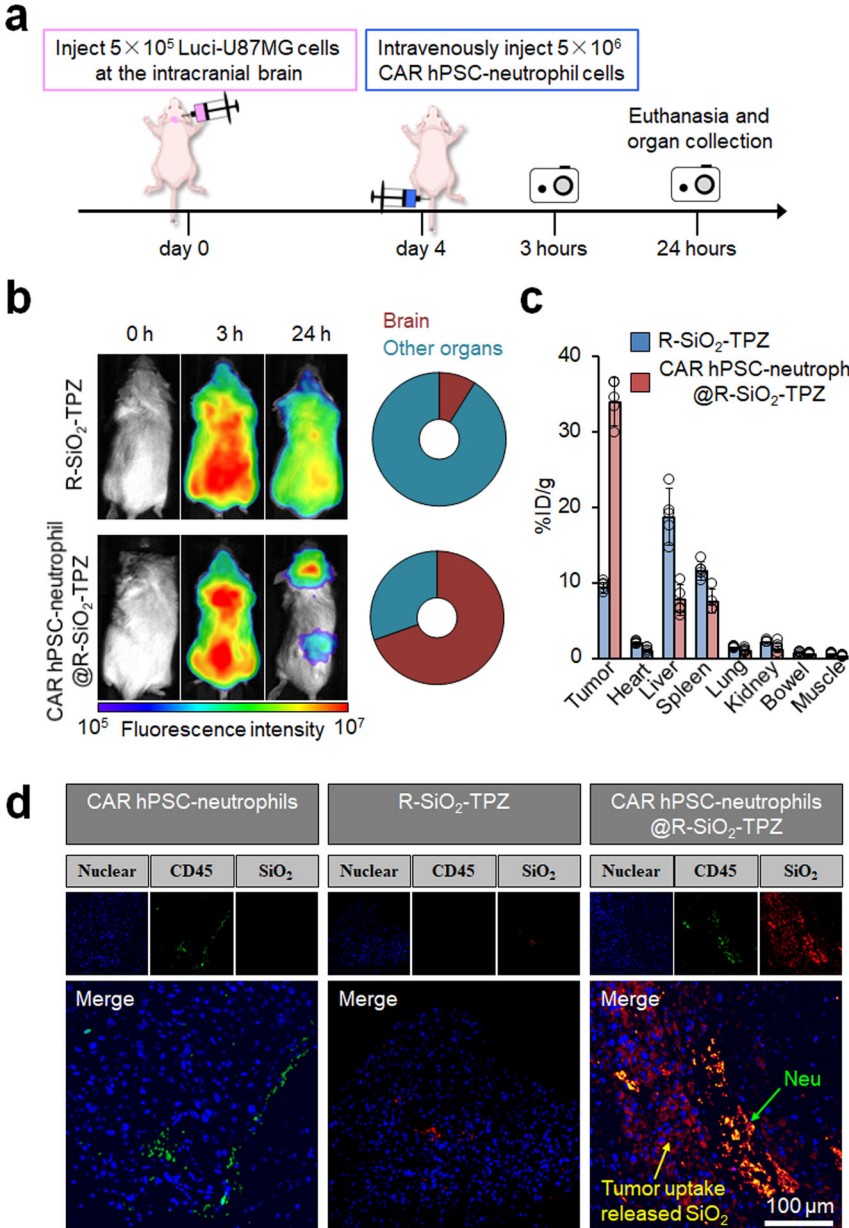

**Fig. 6 | In vivo distribution of CAR neutrophil-delivered R-SiO₂-TPZ nanoparticles (NPs). a** Schematic of intravenously administered Cy5-labeled CAR neutrophil@R-SiO₂ NPs and R-SiO₂ NPs for in vivo cell tracking study. $5 \times 10^5$ luciferase (Luci)-expressing U87MG cells were stereotactically implanted into the right forebrain of NRG mice. After 4 days, mice were intravenously treated with PBS, $5 \times 10^6$ Cy5-labeled CAR neutrophil@R-SiO₂ NPs and R-SiO₂ NPs. **b** Time-dependent biodistribution of Cy5+ neutrophils in whole body, brain, and other organs was determined and quantified by fluorescence imaging at the indicated hours. **c** Biodistribution of CAR neutrophil@R-SiO₂ NPs and R-SiO₂ NPs in mice at 24 h post-injection was analyzed by inductively coupled plasma-optical emission spectrometry (ICP-OES) based on Si element, and data was expressed as the percentage of injected dose per gram of tissue (%ID/g). $n = 5$ biologically independent samples. Data are represented as mean ± SD. Source data are provided as a Source Data file. **d** Representative fluorescence images of CD45 and SiO₂ in the indicated glioblastoma xenografts isolated from tumor-bearing mice were shown. Scale bars, 100 μm. Biological triplicates were performed independently.

neutrophils[29]. We developed a powerful bioinspired neutrophil-mediated drug delivery system with CAR-engineering[29] and used engineered human CAR-neutrophils as a nanocarrier with striking antitumor activities. Rough SiO₂ NPs works better than smooth SiO₂ NPs in CAR-neutrophil carriers, consistent with previous observation that neutrophils preferentially phagocytose rough microbial pathogens[30]. Neutrophils were reported to promote the proliferation and progression of glioma cells[48]. We observed a similar pro-tumor effect of unmodified neutrophils in our animal study, highlighting the necessity of CAR-engineering or other modifications in neutrophils to ensure their safety in drug delivery and other therapeutic applications. Notably, our CAR-neutrophil-mediated drug delivery depends solely on the native chemo-attractant ability of GBM, but not the amplified post-surgical inflammatory signals, suggesting the high specificity and therapeutic potential of our drug-delivery system in eradicating deeply infiltrated gliomas that cannot be removed by surgery. Since surgical resection and adjuvant chemotherapy/radiotherapy are the primary clinical intervention for GBM[12], combination treatment with CAR-neutrophil nanocarriers and surgery/radiotherapy may achieve optimal therapeutic efficacy and is worth further investigation.

T and NK cell-specific CAR constructs have been widely used to enhance anti-tumor activities of T and NK cells, but neutrophil-specific CARs that improve anti-tumor functions of neutrophils have not been described. CD4ζ and CD4γ chimeric immune receptors were

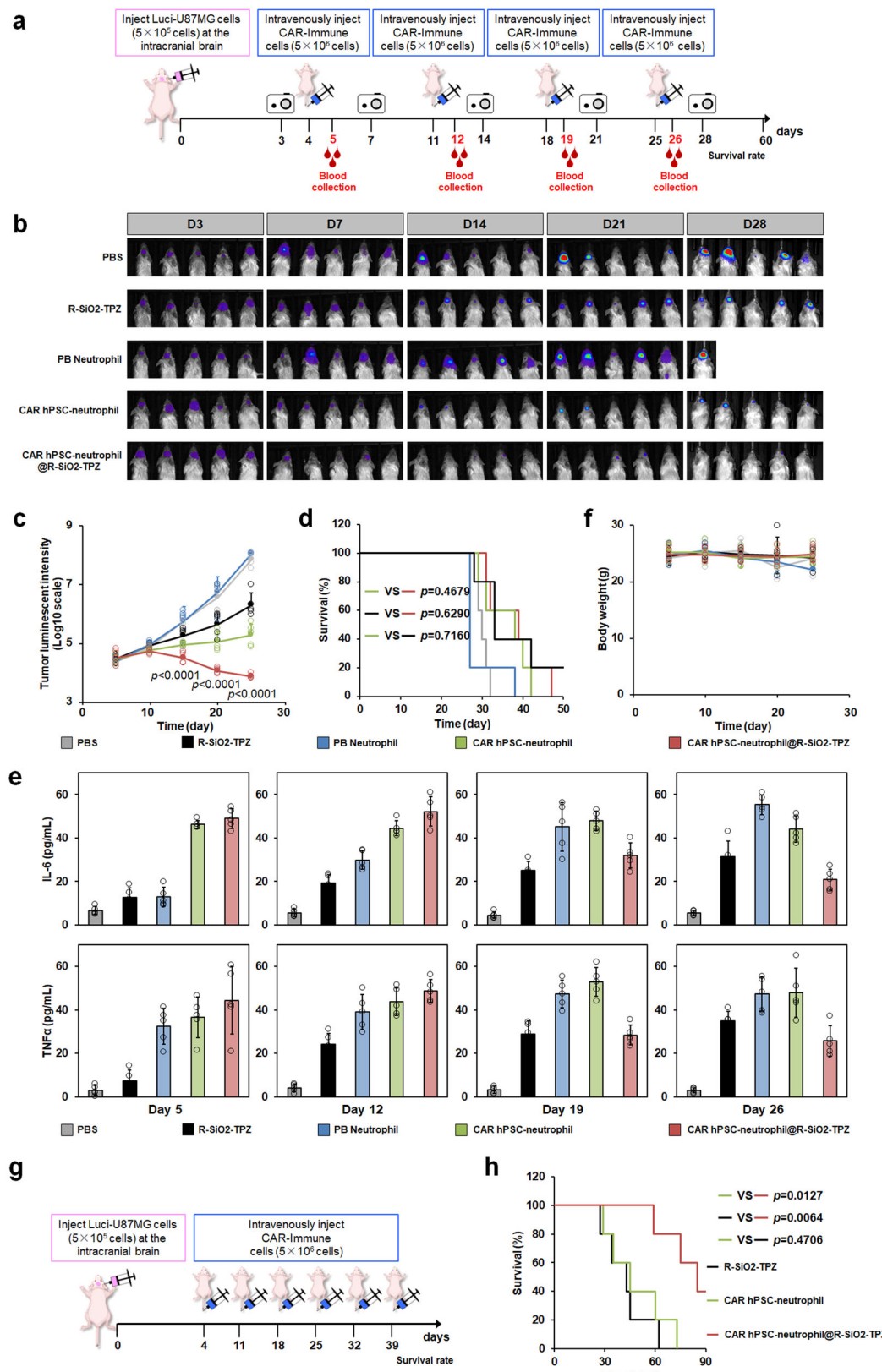

previously reported to enhance the cytolysis of neutrophils against HIVenv-transfected cells in vitro. Still, the lysis efficiency was only ~10% at an effector-to-target (E:T) ratio of 10:1[28]. FcγRIIA (CD32a) is a low-affinity single-chain transmembrane receptor for monomeric IgG that is highly expressed in neutrophils (30,000 to 60,000 molecules/cell[31]), and its ligation induces Fcγ-dependent functions in neutrophils, such

as the release of granule contents, Ca$^{2+}$ mobilization, anti-tumor cytotoxicity, and phagocytosis[49]. Given the prominent role of CD32a in the activation and function of neutrophils, we designed and tested CD32a-based CAR constructs. However, our results demonstrated that CD3ζ mediates significantly better cytolysis than CD32aγ when expressed in hPSC-derived neutrophils, which may be in part due to

**Fig. 7 | In vivo anti-tumor activities of combinatory CAR-neutrophils and R-SiO₂-TPZ nanoparticles (NPs) were assessed via intravenous injection.**
**a** Schematic of intravenously administered PBS, PB-neutrophils, CAR-neutrophils, and CAR-neutrophil@ R-SiO₂-TPZ NPs for in vivo tumor-killing study. $5 \times 10^5$ luciferase (Luci)-expressing U87MG cells were stereotactically implanted into the right forebrain of NRG mice. After 4 days, mice were intravenously treated with indicated neutrophils weekly for a month. Time-dependent tumor burden was determined (**b**) and quantified (**c**) by bioluminescent imaging (BLI) at the indicated days. Data are mean ± SD for mice in (**b**) ($n = 5$), one-way analysis of variance (ANOVA). **d** Kaplan-Meier curve demonstrating survival of indicated experimental groups

($n = 5$) was shown. Released human tumor necrosis factor-α (TNFα) and IL-6 in the peripheral blood (**e**) and body weight (**f**) of different mouse groups were measured at the indicated days. Data are mean ± SD, $n = 5$ biologically independent samples. **g, h** Anti-tumor activity of increased dosage frequencies of CAR-neutrophils and R-SiO2-TPZ NPs was assessed. **g** Schematic of intravenously administered CAR-neutrophils, R-SiO₂-TPZ NPs and CAR-neutrophil@ R-SiO₂-TPZ NPs for in vivo tumor-killing study. **h** Kaplan-Meier curve demonstrating survival of indicated experimental groups was shown ($n = 5$). Kaplan–Meier curves were analyzed by the log-rank test. Source data are provided as a Source Data file.

the higher copies of ITAMs in CD3ζ than CD32aγ: three and one copies, respectively, and higher expression levels of ζ than γ on the cell surface of neutrophils[28]. Like CD32a, FcγRIII (CD16b) is another low-affinity receptor for monomeric IgG and expressed at a much higher level than CD32a on neutrophils[31]. While cross-linking of CD16b only induces $Ca^{2+}$ mobilization and degranulation, but not phagocytosis and cytolysis in neutrophils[28,50], it will still be of interest in future studies to perform a systematic comparison on the abilities of CD3ζ- and CD16bγ-CARs in triggering and enhancing anti-tumor functions of neutrophils.

We also presented here a modular and versatile hPSC-neutrophil drug delivery platform that may be re-engineered and tuned in the future to support other neutrophil-based efforts to treat other human diseases. First, CAR engineering is more accessible in hPSCs than in primary immune T cells and neutrophils. It only requires one-time genome editing to achieve stable and homogenous expression of various CARs[29]. In addition to the CLTX-CARs, we also constructed stable hPSC lines that express a universal anti-fluorescein (FITC)[51] or anti-PD-L1 CAR[52], both of which could be harnessed to obtain universal solid tumor-targeting nanocarrier CAR-neutrophils. Other genetic modifications, such as fibrosis targeting anti-FAP CARs[53], can also be performed to direct neutrophil nanocarriers to treat fatal regenerative diseases, including brain trauma and cardiac fibrosis. Furthermore, CAR-expressing hPSCs could also be easily adapted to produce CAR-T or CAR-NK cells[29], and combinations of these immunotherapies with CAR-neutrophil nanocarriers may achieve optimal therapeutic anti-tumor benefits. Finally, our bioinspired tumor glutathione (GSH)-responsive nanodrug system is a modular and versatile platform to load promising chemotherapeutic or radioactive drugs into CAR-neutrophils for targeted drug delivery, as exemplified by clinical TMZ, JNJ64619187, and pro-drug TPZ. Future studies on testing other nanoparticles may yield optimized drug loading in neutrophils and achieve maximum in vivo therapeutic efficacy.

While we have demonstrated the therapeutic concept of using CAR-neutrophils to specifically and efficiently deliver chemo-drugs to brain tumor across BBB, there are a few limitations in this study. First, 4-day tumor cell inoculation may not be sufficient to establish tumors that mimic the clinical scenario for therapeutic investigation, and future work with different tumor inoculation periods are necessary to recapitulate the different stages of glioblastoma development and therapeutic response in different patients[54,55]. Second, the immunodeficient mice we used here lack adaptive immunity and other preclinical models with an intact immune system, such as pet dogs with spontaneous glioma[56], are needed to better assess the safety and efficacy of CAR-neutrophils produced in vitro. Particularly, off-target toxicity profiling of CAR-neutrophils with or without loading nanodrugs in infused animals, including cytokine release syndrome, neurotoxicity, and on-target off-tumor toxicities observed in CAR-T cells[57], are needed, despite the short lifespan of neutrophils. While feasible approaches, such as engineering hypoimmunogenic universal donor hPSCs[58–61] and banking human leukocyte antigen (HLA)-homozygous hPSC libraries[62], are readily available to avoid the potential risk of graft-versus-host disease (GvHD), preclinical animal models with an intact immune

system is still needed to assess the translational potential of our neutrophil therapeutics. Finally, limited anti-tumor cytotoxicity and extension of animal lifespan of CAR-neutrophil nanodrug therapeutics were observed. Therefore, future exploration of more effective chemotherapy drugs or radiosensitizers, and combinatory therapies with classic CAR-T and surgical resection is essential to achieve maximum anti-tumor efficacy of CAR-neutrophil therapeutics. For instance, a recent study on mechanism-based design has led to a more effective drug KL-50 that overcomes acquired resistance as observed in clinical TMZ drug[63] and can thus be incorporated into our modular CAR-neutrophil nanodrug platform for a potentially better therapeutic efficacy. Extending the shelf life of neutrophils to 5 days via CLON-G (caspases-lysosomal membrane permeabilization-oxidant-necroptosis inhibition plus granulocyte colony-stimulating factor[64]) treatment and/or using a longer-term controlled drug release system in CAR-neutrophils may also achieve sustained in vivo anti-tumor efficacy after neutrophil apoptosis.

Collectively, our findings clearly demonstrated that R-SiO₂−TPZ-loaded CAR-neutrophils could sustain the anti-tumor N1 phenotype and efficiently kill tumor cells under various tumor niche-like conditions in vitro. Functional CAR-neutrophils could also be produced in large quantities from engineered hPSCs to precisely deliver tumor microenvironment-responsive nanodrugs to target GBM in vivo, leading to a combinatory chemoimmunotherapy with robust and specific anti-GBM activities and minimal off-target drug delivery that prolonged lifespan in tumor-bearing mice.

## Methods
### Ethics statement
This study protocol was reviewed and approved by the Purdue University Institutional Biosafety Committee (IBC). All mouse experiments were approved by the Purdue Institutional Animal Care and Use Committee (PACUC) and performed in accordance with the Guide for Care and Use of Laboratory Animals.

### Donor plasmid construction
The donor plasmids targeting *AAVS1* locus were constructed as previously described[65]. Briefly, chlorotoxin (CLTX) sequence[27] containing a GM-CSFR signal peptide and IgG4 hinge, and CD3ζ and/or CD32aγ with CD4 or CD32a transmembrane domain were directly synthesized (Genewiz) and cloned into the AAVS1-Puro CAG-FUCCI donor plasmid (Addgene #136934). The resulting CLTX CAR constructs were sequenced and submitted to Addgene (#171963 to #171965).

### Maintenance and differentiation of hPSCs
mTeSR plus medium was used to culture and maintain H9 hPSCs (WiCell, WA09) on Matrigel-coated plates. Neutrophil differentiation was performed in an iMatrix 511-coated 24-well plate by dissociating and seeding 10,000 and 80,000 cells/cm² hPSCs in mTeSR plus medium containing 5 μM Y27632 (day −1). 6 μM CHIR99021 (CHIR) was used to initiate neutrophil differentiation in DMEM medium supplemented with 100 μg/mL ascorbic acid (DMEM/Vc) at day 0. From day 1 to day 4, LaSR basal medium with 50 ng/mL VEGF (only day 2 to day 4) was used to induce endothelial cell specification[66]. On day 4, 10 μM

SB431542, 25 ng/mL SCF and 25 ng/ml FLT3L were used to promote the endothelial-to-hematopoietic transition in Stemline II (Sigma) medium. On day 6, SB431542 was removed via medium change with Stemline II medium containing 50 ng/mL SCF and 50 ng/mL FLT3L. Half medium change was performed on day 9 and day 12 with 0.5 ml fresh Stemline II medium containing 25 ng/mL GM-CSF, 50 ng/mL SCF, and 50 ng/mL FLT3L. On day 15, floating hematopoietic progenitor cells were harvested by filtering through a cell strainer and differentiated into neutrophils in Stemline II medium containing 1x GlutaMAX, 2.5 µM AM580 and 150 ng/mL G-CSF with half medium change every 3 days. As early as day 21, mature neutrophils could be harvested for in vitro and in vivo applications.

## Flow cytometry analysis

Cell cultures were gently pipetted and filtered through a 70 or 100 µm strainer sitting on a 50 mL tube. The cells were then pelleted by centrifugation and washed twice with PBS−/− solution containing 1% bovine serum albumin (BSA). The cells were stained with appropriate conjugated antibodies (Supplementary Table 1) for 25 min at room temperature in dark, and analyzed in an Accuri C6 plus cytometer (Beckton Dickinson) after washing with BSA-containing PBS−/− solution. FlowJo software was used for flow data analysis.

## Nucleofection and genotyping

Pre-treatment with 10 µM Y27632 for 3–4 h or overnight before nucleofection could be used to increase cell viability of hPSCs. 6 µg SpCas9 AAVS1 gRNA T2 (Addgene #79888) and 6 µg CAR donor plasmids were nucleofected into $1–2.5 \times 10^6$ singularized hPSCs in 100 µl nucleofection solution (Lonza #VAPH-5012) using a Nucleofector 2b device (program code B-016). 3 ml pre-warmed mTeSR plus and 10 µM Y27632 were used to culture the resulting nucleofected hPSCs in one well of a Matrigel-coated 6-well plate, followed by medium change with fresh mTeSR plus containing 5 µM Y27632 after 24 h. Daily medium change was then performed until hPSCs reached 80% confluence, followed by the drug selection with 1 µg/ml puromycin (Puro) for about 1 week. When nickel-sized hPSC clones were visible, a EVOS XL Core microscope (ThermoFisher) was used to pick individual clones into each well of a 96-well plate pre-coated with Matrigel. After expansion for 2–5 days, the genomic DNA of single clone-derived hPSCs was extracted with QuickExtract™ DNA Extraction Solution (Epicentre #QE09050) and subjected for PCR genotyping using 2×GoTaq Green Master Mix (Promega #7123). The primer pairs used for positive and homozygous genotyping were listed in Supplementary Table 2.

## Synthesis of degradable dendritic mesoporous organosilica nanoparticles (DDMONs)

DDMONs were synthesized via a one-pot synthesis using NaSal and cationic surfactant CTAB as structure directing agents, tetraethyl orthosilicate (TEOS) and bis[3-(triethoxysilyl)propyl] tetrasulfide (BTES) as silica source, and triethanolamine (TEA) as a catalyst. The synthesis was conducted in a 50 mL flat bottom glass bottle with a 3-cm stirring bar. Typically, 0.034 g of TEA was added to 12.5 mL of water and stirred gently (~700 rpm) at 80 °C in an oil bath under a magnetic stirrer for 0.5 h. Afterward, 190 mg of CTAB and 42 mg of NaSal were added to the above solution, stirring for another 1 h. After CTAB and NaSal were completely dissolved, a mixture of 1 mL of TEOS and 0.8 mL of BTES was added to the mixture solution, followed by vigorous stirring for 12 h. The nanoparticles were collected by centrifugation at 20, 000 rpm for 5 min and washed three times with ethanol to remove residual reactants. The powder was then dried in a vacuum oven at 40 °C for 6 h. Collected products were extracted with HCl and methanol solution at 60 °C for 6 h three times to remove the template, followed by overnight vacuum drying at room temperature.

## Preparation of sphere mesoporous silica nanoparticles (SMSNs)

In order to prepare the SMSNs, 240 mL of aqueous solution containing 0.5 g of CTAB was prepared in a conical flask. 1.5 mL of NaOH (2 mol L$^{-1}$) was then added to the CTAB solution with stirring for 10 min. Once the mixture temperature was adjusted to 80 °C, 2.5 mL of TEOS was added dropwise to the solution and stirred for 2 h. The resulting SMSNs were then centrifuged and washed with ethanol and deionized water several times to remove surfactant templates. The remaining powder was dried in a vacuum oven at 40 °C for 6 h. The collected products were extracted with HCl and methanol solution at 60 °C for 6 h three times to remove residual template, followed by overnight vacuum drying at room temperature.

## Drug loading and glutathione-stimulated release

For tirapazamine (TPZ) loading, 5 mL of 1 mg mL$^{-1}$ TPZ in phosphate buffer solution was mixed with 5 mL of SiO$_2$ suspension (5 mg mL$^{-1}$) in phosphate buffer solution (20 mM, pH = 7.4), and the resulting solution was continuously stirred at 37 °C for different time points (0.5, 1, 2, 4, 6, 12, and 24 h). Unloaded TPZ was removed by centrifugation at 8000 rpm for 10 min and the pellet was washed with phosphate buffer solution for three times. TPZ in the supernatant was determined by UV-Vis spectroscopy. TPZ loading capacity (LC) was calculated as following: LC = (total TPZ − TPZ in supernatant)/(total TPZ) ×100%. For glutathione (GSH)-stimulated release analysis, 10 mL of TPZ@SiO$_2$ suspension in phosphate buffer solution (1 mg mL$^{-1}$) were incubated with 10 mM GSH at different time points (10, 20, 30, 40, 50, and 60 h). 1 mL of the complex dispersion was removed and centrifuged at 8000 rpm for 10 min, and TPZ released to the supernatant was quantified by UV-Vis spectroscopy. Similar procedures were performed to load TMZ and JNJ64619187.

## Loading nanodrugs into neutrophils

Nanodrug loading was prepared by incubating neutrophils with TPZ@SiO$_2$ or nanoparticles loaded with TMZ or JNJ64619187. Briefly, hPSC-derived neutrophils ($1 \times 10^5$ cells/mL) were placed in a DNA low-bind tube and incubated with nanodrugs (equivalent to 400 µg/mL of SiO$_2$ content) for 1 h. After centrifugation and PBS washing three times, nanodrug/neutrophils were resuspended in PBS and ready for subsequent experiments. Uptaking efficiency of nanodrugs by neutrophils was measured by flow cytometry, and the location of nanodrugs within neutrophils was determined by a fluorescence microscope. Neutrophil viability after incubating with nanodrugs for 4 and 8 h was measured by Zombie Green Fixable Viability Kit (BioLegend). In order to quantify the loading content of SiO$_2$, nanodrug/neutrophil samples were digested by tetramethylammonium hydroxide and a high pressure, and the silicon concentrations of digested samples were measured by inductively coupled axial plasma optical emission spectrometry (ICP-OES).

## Transwell migration assay

After resuspended in HBSS buffer, neutrophils were placed on the top chamber of a transwell and allowed to migrate towards 10 and 100 nM fMLP in the bottom chamber. After 2 h, neutrophils in the bottom chamber were collected and quantified in an Accuri C6 plus cytometer (Beckton Dickinson). FlowJo software was then used to quantify the migrated live neutrophils, which were normalized by the total number of seeded neutrophils.

## 2D chemotaxis assay

After resuspended in HBBS buffer containing 0.5% FBS and 20 mM HEPES, neutrophils were loaded into IBIDI chemotaxis µ-slides pre-coated with collagen, and allowed to attach for 30 min at 37 °C. In the presence of 187 nM fMLP (by adding 15 µL of 1000 nM fMLP to the right reservoir), neutrophil migration was recorded in a Zeiss LSM 710 Confocal Microscope with Ziess EC Plan-NEOFLUAR

10X/0.3 objective every minute for a total of 120 min at 37 °C, and the resulting cell migration video was analyzed with ImageJ plug-in MTrackJ.

## Neutrophil-mediated in vitro cytotoxicity assay

For cytotoxicity analysis, both live/dead cell staining and CytoTox-Glo™ Cytotoxicity Assay kit (Promega) were used. Briefly, 5000 tumor cells (100 μL) and 15,000, 25,000 or 50,000 neutrophils (100 μL) were mixed and incubated in the 96-well plate for 24 h at 37 °C. Floating cells were first collected by transferring cell-containing media into a new round-bottom 96-well plate. Attached cells were then dissociated by 50 μL trypsin-EDTA and added into the same wells with floating cells, followed by centrifugation at $300 \times g$, 4 °C for 4 min. For flow cytometry analysis, pelleted cells were washed with 200 μL 0.5% BSA-containing PBS−/− solution, stained with CD45 antibody/Calcein AM for half an hour, and analyzed in an Accuri C6 plus cytometer (Beckton Dickinson). CytoTox-Glo™ cytotoxicity analysis and quantification were determined by SpectraMax iD3 microplate reader (Molecular Devices, Sunnyvale, CA, USA).

## Conjugate formation assay

To visualize immunological synapses, 200 μL of U87MG cells (50,000 cells/mL) were seeded onto wells of a 24-well plate and incubated at 37 °C for 12 h to allow cells to attach. 200 μL neutrophils (500,000 cells/mL) were then added onto the target U87MG cells and incubated for 6 h before fixation with 4% paraformaldehyde (in PBS). Cytoskeleton staining was then performed using an F-actin Visualization Biochem Kit (Cytoskeleton Inc.). To quantify the immunological synapses, a total of $1 \times 10^6$ CAR hPSC-neutrophils or CAR hPSC-neutrophils@RSiO$_2$-TPZ labeled with anti-CD45-APC (BD Biosciences) were incubated with $2 \times 10^5$ targeted cells stained with Calcein-AM fluorescent dye (Invitrogen) for various time points at 37 °C and humidified 5% $CO_2$ atmosphere. After incubation, the cells were fixed and analyzed in an Accuri C6 plus cytometer (Beckton Dickinson) after washing with BSA-containing PBS−/− solution, and cells of double-positive events APC + / Calcein+ were analyzed to quantify immunological synapse formation in FlowJo software.

## Measurement of reactive oxygen species (ROS) production

In total, 12 h after seeding 3000 U87MG tumor cells (100 μL) into wells of a 96-well plate, 30,000 neutrophils were added into the same wells at a 10:1 neutrophil-to-tumor ratio. In total, 12 h after co-incubation at 37 °C, 10 μM H$_2$DCFDA was added to the cell mixture and incubated for 50 min. The ROS production was measured in a SpectraMax iD3 microplate reader (Molecular Devices, Sunnyvale, CA, USA) by quantifying the fluorescence emission signal (480–600 nm) with an excitation wavelength of 475 nm.

## ELISA analysis of TNFα level

In total, 100 μL of U87MG cells (30,000 cells/mL) were seeded into wells of a 96-well plate 12 h before adding neutrophils at a neutrophil-to-tumor ratio of 10:1. After co-incubation for 12 h, the plate was centrifuged at 2000 rpm for 10 min to spin down the cell debris, and 50 μL of supernatant was transferred for measurement of TNFα activity using TNFα ELISA kit (Thermo Fisher, Cat#BMS223-4) according to the manufacturer's instructions. A standard cytokine dilution series or 50 μL of supernatant were pipetted into the pre-coated wells for antigen capture. After 2 h, the plate was washed with buffer solution five times to remove the unbound factors, and an enzyme-linked secondary polyclonal antibody was added for 30 min of incubation. Following the washing process, a substrate solution was added and incubated for 30 min. After termination of the reaction by adding 50 μL of stop buffer solution, the colorimetric intensity was measured at 450 nm by a SpectraMax iD3 microplate reader (Molecular Devices, Sunnyvale, CA, USA).

## Blood-brain-barrier (BBB) transmigration assay

HBEC-5i cells were used to establish the in vitro BBB model using Transwell culture plates. Briefly, the transwell inserts of a Corning 24-well Transwell plate (8 μm pore size, 6.5 mm diameter) were pre-coated with 1% gelatin (w:v) and seeded with $1 \times 10^5$ HBEC-5i cells (cells/well) in 10% FBS-containing DMEM/F12 medium. The upper chamber of the resulting Transwell BBB model was then loaded with $2 \times 10^5$ neutrophils, and the lower chamber was filled with FBS-free medium in the presence or absence of 10 nM fMLP. 3 h later, neutrophil-containing media in the upper and lower chambers were then collected for neutrophil quantification. For cytotoxicity analysis, the lower chamber was seeded with $2 \times 10^4$ U87MG tumor cells and 12 h later, the upper chamber was then seeded with $2 \times 10^5$ neutrophils in FBS-free medium with 10 nM fMLP. After incubated for 12 h, tumor cells were collected and subjected for cell viability analysis using a flow cytometer. For the second migration analysis, the upper chamber of the second Transwell plate was loaded with $2 \times 10^5$ neutrophils from the bottom chamber of the first BBB model, and the number of neutrophils migrated towards the bottom chamber with tumor cells was measured.

## Neutrophil infiltration of 3D tumor spheroids

The hanging-drop method was used to generate 3D tumor spheroids. Briefly, 20 μL of $2 \times 10^6$ cells/mL U87MG cells in MEM medium containing 10% FBS and 0.3% methylcellulose were pipetted and seeded as an individual droplet onto an inverted lid of a 96-well plate. PBS was used to fill the wells of the 96-well plate and the lid with hanging droplets was then returned to cover the plate. After 5–7 days of incubation at 37 °C and 5% $CO_2$, cell aggregates were formed and transferred to a 24-well plate with one aggregate per well. $2 \times 10^5$ neutrophils were then added to each well and co-cultured with the tumor spheroid for tumor penetration assessment. 24 h later, tumor spheroids with infiltrating neutrophils were fixed, followed by CD45 and DAPI staining. Live/dead cell staining using flow cytometry analysis or CytoTox-Glo™ Cytotoxicity Assay using a SpectraMax iD3 microplate reader were performed to measure tumor cell cytotoxicity.

## Bulk RNA sequencing and data analysis

Bulk RNA sequencing (RNA-seq) was performed according to our previous study[29]. Briefly, neutrophil and U87MG cell cocultures were washed with PBS for five times to remove residual floating neutrophils, and attached U87MG cells were dissociated with 0.25% Trypsin-EDTA solution for total RNA isolation using Direct-zol RNA MiniPrep Plus kit. RNA samples were then prepared and sequenced in Illumina HiSeq 2500 by the Center for Medical Genomics at Indiana University. The resulting sequencing reads were mapped to the human genome (hg19) using HISAT2 program, and the RefSeq transcript levels (RPKMs) were then quantified using the python script rpkmforgenes.py. Heatmaps of selected gene subsets after normalization were plotted using Morpheus (Broad Institute). The original fastq files and processed RPKM text files were available in NCBI with the accession number GSE206170 (https://www.ncbi.nlm.nih.gov/geo/query/acc.cgi?acc=GSE206170).

## Mouse xenograft studies

All mouse experiments were approved by the Purdue Institutional Animal Care and Use Committee (PACUC). The immunodeficient NOD.Cg-$RAG^{1tm1Mom}IL2rg^{tm1Wjl}$/SzJ (NRG) mice were bred and maintained by the Biological Evaluation Core at Purdue University Institute for Cancer Research. All the female mice used in this study were 6- to 10-week-old. Mice were housed in pathogen free and ventilated cages, and allowed free access to autoclaved food and water, in a 12-h light/dark cycle, with room temperature at $21 \pm 2$ degree and humidity between 45 and 65%. In situ xenograft murine models were constructed via intracranial injection of $5 \times 10^5$ luciferase-expressing GBM cells into the brain of immunodeficient mice. $5 \times 10^6$ neutrophils were intravenously injected at day 4, day 11, day 18, and day 25, and blood

samples were collected from these mice at day 5, day 12, day 19, and day 26. Tumor burden was monitored by bioluminescence imaging (BLI) system (Spectral Ami Optical Imaging System) and body weights of experimental mice were measured once per week. Collected blood cells were stained with CD45 and analyzed in an Accuri C6 plus flow cytometer (Beckton Dickinson). Blood samples were also subjected for enzyme-linked immunosorbent assay (ELISA) to measure human TNFα and IL-6 cytokine release (Invitrogen). At the end of treatment, tumors were collected for H&E staining. The experimental endpoint was defined as death, luciferase signal intensity in bioluminescence imaging higher than $10^9$ a.u., or experiencing neurological symptoms (*i.e.*, inactivity, seizure, ataxia, and/or hydrocephalus). The mice bearing tumor over $10^9$ a.u. or experiencing neurological symptoms were euthanized. For in vivo biodistribution analysis, fluorescence images were captured by the Spectral Ami Optical Imaging System 3 and 24 h after intravenous injection of Cy5 (Lumiprobe)-labeled neutrophils.

## Statistical analysis
Data are presented as mean ± standard deviation (SD). Statistical significance was determined by Student's *t* test (two-tail) between two groups, and three or more groups were analyzed by one-way analysis of variance (ANOVA) in Excel. $p < 0.05$ was considered statistically significant.

## Reporting summary
Further information on research design is available in the Nature Portfolio Reporting Summary linked to this article.

## Data availability
RNA-sequencing raw data and processed files have been deposited and are publicly available on Gene Expression Omnibus (GEO) under accession number GSE206170. The remaining data are available within the Article, Supplementary Information or Source Data file. Source data are provided with this paper.

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

## Acknowledgements

We thank members of the Bao laboratory for their technical assistance. We also gratefully acknowledge the Purdue Flow Cytometry and Cell Separation Facility, Purdue Genomics Core Facility and the Biological Evaluation Core at Purdue University Center for Cancer Research (PCCR). This study was supported by startup funding from the Davidson School of Chemical Engineering and the College of Engineering at Purdue (X.B.), PCCR Robbers New Investigators (X.B.), Purdue Libraries Open Access Fund (X.B.), NIH NIGMS (grant no. R35GM119787 to Q.D.), and NIH NCI (grant no. R37CA265926 to X.B.). The authors also gratefully acknowledge support from the Purdue University Institute for Cancer Research, P30CA023168, Purdue Institute for Integrative Neuroscience (PIIN) and Bindley Biosciences Center, and the Walther Cancer Foundation.

## Author contributions

Y.C. and X.B. conceived and designed the experiments. Y.C. and X.C. synthesized and characterized the SiO$_2$ nanoparticles. R.S., Y.Y., Y.X., G.J., and V.J.B. contributed to design and assisted in data collection and analysis. Y.C., S.T.-A, and B.D.E. designed and performed in vivo experiments. Y.W., Q.D., X.L.L., X.W., and O.E.A. provided critical resources and technical assistance. All authors contributed to the

interpretation of experiments. Y.C. and X.B. wrote the manuscript with support from all authors.

## Competing interests

Y.C., R.S., Q.D., and X.B. are inventors on two patent applications (human chimeric antigen receptor neutrophils, compositions, kits and methods of use (WO2022125850A1), and blood-brain barrier-penetrating CAR-neutrophil-mediated drug delivery (provisional patent)) for content described in this manuscript. X.B. is an advisory chief scientific officer (CSO) for Astheneia Bio. The remaining authors declare no other competing interests.

## Additional information

[1]Davidson School of Chemical Engineering, Purdue University, West Lafayette, IN 47907, USA. [2]Purdue University Institute for Cancer Research, West Lafayette, IN 47907, USA. [3]Tongji University Cancer Center, Shanghai Tenth People's Hospital, Tongji University School of Medicine, Shanghai 200072, China. [4]Department of Biological Sciences, Purdue University, West Lafayette, IN 47907, USA. [5]Division of Chemistry and Chemical Engineering, California Institute of Technology, Pasadena, CA 91125, USA. [6]William G. Lowrie Department of Chemical and Biomolecular Engineering, The Ohio State University, Columbus, OH 43210, USA. [7]Department of Chemical Engineering, Imperial College London, South Kensington Campus, London SW7 2AZ, UK. [8]Department of Comparative Pathobiology, Purdue University, West Lafayette, IN 47907, USA. [9]Department of Biomedical Engineering, The Pennsylvania State University, University Park, PA 16802, USA. [10]The Huck Institutes of the Life Sciences, The Pennsylvania State University, University Park, PA 16802, USA. [11]Department of Biology, The Pennsylvania State University, University Park, PA 16802, USA. [12]Sustainability Institute, The Ohio State University, Columbus, OH 43210, USA. [13]Department of Chemical Engineering, University of Michigan, Ann Arbor, MI 48109, USA. ✉e-mail: lian@psu.edu; wang.12206@osu.edu; lolaa@umich.edu; bao61@purdue.edu

