## [Peer Review File · Nature Communications]

CAR-neutrophil mediated delivery of tumor- microenvironment responsive nanodrugs for glioblastoma chemoimmunotherapyEditorial Notes:

This manuscript has been previously reviewed at another journal that is not operating a transparent peer review scheme. This document only contains reviewer comments and rebuttal letters for versions considered at *Nature Communications*.

REVIEWER COMMENTS

Reviewer #1 (Remarks to the Author):

The revised manuscript by Chang and colleagues is much improved and they were able to adequately address most of the comments. Still, several issues remain to be clarified before the manuscript can be further considered for publication. [redacted to maintain the confidentiality of unpublished data.]

Major comments:

1. Figure S3A and B show ROS and TNF α production in the indicated neutrophil groups following incubation with SVG p12 cells. If this is indeed the case, what is the data show in the leftmost bar (SVG p12 only)? Does this mean that the SVG p12 cells cultured alone generate as much ROS and TNF α as the neutrophils in coculture?
2. High background killing of tumor cells by neutrophils (60%) overshadows the real/specific effect of the modified neutrophils. Instead of using a 10:1 ratio in figure S3A the authors should use 5:1 or 1:1 and show that while glioblastoma cells are eliminated other cells are spared.
3. The observations made in the response to my comment no. 6 should be added to the revised manuscript.
4. I fail to understand the dramatic difference between the result from the experiment in figure 7A-B and the experiment in figure 7G-H. In 7G-H the amount of neutrophils/dose remained the same with the addition of two more neutrophil doses on days 32 and 39. However, if we look at figure 7D, by day 32 40% of the mice treated with hSPC-neutrophil-R-Sio2-TPZ have already died. How did an additional doses on day 32 and 39 improved this? Although the authors rightfully point to the short life span of neutrophils, the additional doses of neutrophils at late timepoints does not resolve this problem.

[redacted to maintain the confidentiality of unpublished data.]

Minor comments:

1. Figure 7H should show the duration of the experiment until all mice perish.

Reviewer #2 (Remarks to the Author):

The authors tried their best to answer the comments from previous reviewers with multiple additional experiments. Especially, it is clear that they understood our concerns about the translational capability of this treatment in the future, given the potential risk on GvHD and lack of significant survival benefit in vivo. Although the authors tried to overcome one of the limitations of their original CAR-neu system having short lifespan by successive administration of CAR-neu with nanodrugs, the other fundamental question about the source of neutrophil remains. Allogenic approach of this system would later cause critical problems during clinical translation and would offset the therapeutic benefits shown in in vivo models.

[redacted to maintain the confidentiality of unpublished data.]

[redacted to maintain the confidentiality of unpublished data.]

Reviewer #3 (Remarks to the Author):

I think this is a much improved manuscript from the initial submission, with the following comments:

1. My comment from the previous review still stands- I don't know that only looking at 24h viability of the CARneutrophils after NP and drug loading was sufficient when compared to the assays looking at 72h results in vitro. (Unless they assert that the drug can have delayed cytotoxicity up to 48h after release, which should be discussed in the discussion if so). Especially because if these are administered peripherally, there are still some in circulation after 24h that could spontaneously be releasing drug elsewhere upon apoptosis. Especially if neutrophils are naturally fragile.

2. Considered the high level of lysis observed in normoxic conditions (which theoretically would be native neutrophil-cytotoxicity and not the hypoxia-activated drug), I think that more validation of CAR neutrophil off-target toxicity should be at least referenced in the discussion (assuming it is in their initial manufacturing paper). [redacted to maintain the confidentiality of unpublished data.]

3. I would have liked to see overlap between CAR + and NP-loaded neutrophils, since drug loading was apparently around 10% of the neutrophils. What percent were double positive? Do you think this population did the bulk of the work here?

[redacted to maintain the confidentiality of unpublished data.]

5. It looks like the bulk RNA sequences methods were left out of the methods section (Figure 4)- I was looking to see if the authors had any information on their bulk RNA seq data. I am curious as to what measures they took in the pipeline to ensure that only tumor cell RNA data was used out of those co-cultures, because I'm wondering if some of their hits are from dying neutrophils instead of the tumor cells.

6. There appears to be a labeling error in Figure 7H graph.

7. In Figure 5, they show the CAR neutrophils penetrating GBM spheroids. What would have made this figure more convincing is some IHC proof of a hypoxic gradient created within these spheroids to prove activity of the TPZ release and cytotoxicity.

8. In Figure S7, how were immunological synapses quantified?

Reviewer #4 (Remarks to the Author):

In this manuscript, the authors developed CAR-neutrophils loaded with therapeutic nanoparticles for glioblastoma therapy. Given the resistance of neutrophils to genomic engineering, the authors decided to create CAR-expressing human pluripotent stem cells and then induced their differentiation into CAR-neutrophils. The authors used these CAR-neutrophils as whole cell carriers to encapsulate silica nanoparticles loaded with hypoxia-responsive chemicals for cancer therapy. Through a series of in vitro studies, they demonstrated that the drug-loaded CAR-neutrophils targeted brain tumor cells, released therapeutic cargo in response to hypoxic conditions, and induced cancer killing. In an in vivo mouse glioblastoma model, drug-loaded CAR-neutrophils, following systemic administration, were shown to accumulate in the brain, inducing anti-cancer effects and improving the mouse survival rate. Overall, the concept presented in this work is interesting. The authors have put in considerable effort in addressing all prior comments. However, the revised manuscript, as it stands, still contains some issues, as detailed in the comments below.

1. Using whole cells as carriers to deliver drug-loaded nanoparticles is not as straightforward as what may have been suggested in this manuscript. The authors did not provide sufficient information on how to load nanoparticles into cells. For example, did the authors try different nanoparticle input to optimize loading yield? Did they optimize the length of time for co-incubating cells and nanoparticles? Also, following the cellular uptake, nanoparticles may undergo degradation inside the cells. Did the authors investigate the kinetics of intracellular degradation of nanoparticles?
2. In the introduction section, the authors made some questionable statements. For example, the authors claimed that "Neutrophils' innate immunity and plasticity against various pathogens and cancers, including GBM, were not previously explored in drug delivery systems". This is plainly inaccurate, as many neutrophil-based therapeutic modalities have emerged for treating various diseases including glioblastoma.
3. In the in vitro tumor penetration study, are the images shown in Figure 5I the cross sections of tumor spheroids? Did the authors quantify the penetration depth of drug-loaded CAR-neutrophils? Also, the authors state that "R-SiO₂-TPZ NPs were encapsulated stably in the CAR-neutrophils during tumor infiltration". This statement is not convincing, as neutrophils can theoretically interact with tumor cells and undergo cytolysis at any point during tumor penetration.
4. The authors added a new therapeutic efficacy study in Figures 7G-H in the revised manuscript and noted considerably improved efficacy compared with their previous study in Figure 7. It seems that the only difference between the old and new study is the total number of doses was increased from 4 to 6 in the new study. But given that those extra two doses were added on 32 and 39 days, respectively, why did the survival rates show considerable improvement compared with the old study even before 32 days?

Reviewer #1 (Remarks to the Author):

The revised manuscript by Chang and colleagues is much improved and they were able to adequately address most of the comments. Still, several issues remain to be clarified before the manuscript can be further considered for publication. [redacted to maintain the confidentiality of unpublished data.]

Response: We would like to thank the reviewer for the nice summary and appreciation of our previous revised work. We are also grateful for the valuable comments to substantially improve our manuscript, and per the reviewer's suggestion, we have performed additional experiments to address the reviewer's concerns raised from last submission. [redacted to maintain the confidentiality of unpublished data.]

Major comments:

1. Figure S3A and B show ROS and TNF α production in the indicated neutrophil groups following incubation with SVG p12 cells. If this is indeed the case, what is the data show in the leftmost bar (SVG p12 only)? Does this mean that the SVG p12 cells cultured alone generate as much ROS and TNF α as the neutrophils in coculture?

Response: We thank the reviewer for this comment and apologize for the confusing presentation of our data. In **Fig. S3A** and **S3B**, these data were collected from SVG p12 cells alone or SVG p12 cells incubated with different neutrophils to show the low activation of CAR-neutrophils against healthy glial cells in the brain. To make this data clearer, we have included a positive control where U87MG tumor cells were cocultured with the most

effective CAR #1 hPSC-neutrophils. Please see updated data in **Fig. S3A-B** and also figure below.

2. High background killing of tumor cells by neutrophils (60%) overshadows the real/specific effect of the modified neutrophils. Instead of using a 10:1 ratio in figure S3C the authors should use 5:1 or 1:1 and show that while glioblastoma cells are eliminated other cells are spared.

Response: We thank the reviewer for this comment and per the reviewer’s suggestion, we have performed the tumor-killing experiment at a 3:1 ratio of neutrophil:tumor. Please see our updated data in **Fig. S3C** and also figure below.

3. The observations made in the response to my comment no. 6 should be added to the revised manuscript.

Response: We thank the reviewer for this comment, and per the reviewer' suggestion, we have added data in response to last comment no. 6 in **Fig. S5**. We also updated our text and copied below.

Previous studies indicate activation of Syk-Erk signaling pathway leads to the ROS production [42–45]. Therefore, we detected and compared Syk-Erk activation in unmodified neutrophils and CAR-neutrophils, and our results suggested a significantly higher activation of Syk-Erk pathway in CAR-neutrophils under hypoxia (**Fig. S5A-D**), which may sustain the unchanged ROS production of CAR-neutrophils under hypoxia.

4. I fail to understand the dramatic difference between the result from the experiment in figure 7A-B and the experiment in figure 7G-H. In 7G-H the amount of neutrophils/dose remained the same with the addition of two more neutrophil doses on days 32 and 39. However, if we look at figure 7D, by day 32 40% of the mice treated with hSPC-neutrophil-R-SiO₂-TPZ have already died. How did an additional doses on day 32 and 39 improved this? Although the authors rightfully point to the short life span of neutrophils, the additional doses of neutrophils at late timepoints does not resolve this problem.

Response: We thank the reviewer for this comment and apologize for the confusing presentation of our data in previous **Fig. 7**. As pointed out by the reviewer, the short lifespan of neutrophils significantly affects their *in vivo* antitumor efficacy after intravenous injection. Therefore, increased lifespan of neutrophils in the first 4 doses

(before additional doses), as a result of much shorter cell preparation time (in total ~1 hr vs 4 hr) from less experimental groups and less cell isolation during our animal study in **Fig. 7G-H** as compared to **Fig. 7A-D**, increased animal survival in tumor-bearing mice in **Fig. 7G-H**. Furthermore, the additional doses of neutrophils further extended animal survival. While a similar survival curve of R-SiO₂-TPZ group was observed between these two independent animal studies, we'd also like to acknowledge the potential contribution of batch variability to the different survival rate observed in the CAR-neutrophil experimental groups. We have updated our text to make this data clearer and also copied below.

While a similar survival curve of the R-SiO₂-TPZ group was observed between these two independent animal studies, reduced time in cell isolation and preparation for injection from a total of ~4 hrs to 1 hr during the first 4 neutrophil doses led to improved animal survival in CAR-neutrophil groups before day 32. Collectively, our data demonstrated the importance of neutrophil preparation and dosage optimization in future clinical application of neutrophil therapeutics.

[redacted to maintain the confidentiality of unpublished data.]

Minor comments:

1. Figure 7H should show the duration of the experiment until all mice perish.

Response: We thank the reviewer for this comment and per the reviewer's suggestion, we have showed survival curve until all mice perish. Please see updated **Fig. 7H** and also figure below:

Reviewer #2 (Remarks to the Author):

The authors tried their best to answer the comments from previous reviewers with multiple additional experiments. Especially, it is clear that they understood our concerns about the translational capability of this treatment in the future, given the potential risk on GvHD and lack of significant survival benefit in vivo. Although the authors tried to overcome one of the limitations of their original CAR-neu system having short lifespan by successive administration of CAR-neu with nanodrugs, the other fundamental question about the source of neutrophil remains. Allogenic approach of this system would later cause critical problems during clinical translation and would offset the therapeutic benefits shown in in vivo models.

[redacted to maintain the confidentiality of unpublished data.]

[redacted to maintain the confidentiality of unpublished data.]

Response: We would like to thank the reviewer for the nice summary and appreciation of our revised work. We are also grateful for the valuable comments to substantially improve our manuscript. [redacted to maintain the confidentiality of unpublished data.]. As pointed out

by the reviewer, we discussed the translational challenge of allogenic neutrophils in our updated Discussion section and also copied below:

Particularly, off-target toxicity profiling of CAR-neutrophils with or without loading nanodrugs in infused animals, including cytokine release syndrome, neurotoxicity, and on-target off-tumor toxicities observed in CAR-T cells [60], are needed, despite the short lifespan of neutrophils.

Reviewer #3 (Remarks to the Author):

I think this is a much improved manuscript from the initial submission, with the following comments.

Response: We would like to thank the reviewer for the nice summary and appreciation of our revised work. We are also grateful for the valuable comments to substantially improve our manuscript, and we have performed additional experiments to address the reviewer's concerns raised from previous revision.

1. My comment from the previous review still stands- I don't know that only looking at 24h viability of the CAR neutrophils after NP and drug loading was sufficient when compared to the assays looking at 72h results in vitro. (Unless they assert that the drug can have delayed cytotoxicity up to 48h after release, which should be discussed in the discussion if so). Especially because if these are administered peripherally,

there are still some in circulation after 24h that could spontaneously be releasing drug elsewhere upon apoptosis. Especially if neutrophils are naturally fragile.

Response: We thank the reviewer for this comment and apologize for not addressing the reviewer's previous concerns on neutrophil viability and potential off-target drug release upon apoptosis. Consistent with previous studies using mouse neutrophils^{5,6}, ~80% human neutrophils with or without loading nanodrugs die in ~72 hr⁷ (please see **Fig. S6I** and also figure below). We also appreciate the reviewer's concern on the potential off-target drug release upon neutrophil apoptosis, and to minimize or prevent this side effect *in vivo*, we have designed and used the biodegradable mesoporous organic silica nanoparticles to carry therapeutic drugs, which will only release the loaded drugs like TPZ in response to the high cellular GHS within tumor cells^{8,9}. Thus, delayed cytotoxicity up to 48h after drug release was observed *in vitro* using our GHS-responsive nanoparticles, though it is still challenging to fully avoid the off-target drug release and achieve sustained antitumor efficacy. Extending the shelf life of neutrophils and/or using a longer-term controlled drug release system in CAR-neutrophils may lead to minimal off-target drug release upon neutrophil apoptosis and achieve longer-term (beyond 48 hr) antitumor efficacy for future clinical applications. Per the reviewer's suggestion, we discussed the limitation of our CAR-neutrophil mediated drug release in the Discussion section and also copied below.

Fig. S6I. *In vitro* neutrophil viability analysis at indicated time points.

Extending the shelf life of neutrophils to 5 days via CLON-G (caspases-lysosomal membrane permeabilization-oxidant-necroptosis inhibition plus granulocyte colony-stimulating factor [68]) treatment and/or using a longer-term controlled drug release system in CAR-neutrophils may achieve sustained *in vivo* antitumor efficacy after neutrophil apoptosis.

2. Considered the high level of lysis observed in normoxic conditions (which theoretically would be native neutrophil-cytotoxicity and not the hypoxia-activated drug), I think that more validation of CAR neutrophil off-target toxicity should be at least referenced in the discussion (assuming it is in their initial manufacturing paper).

[redacted to maintain the confidentiality of unpublished data.]

Response: We thank the reviewer for this comment, and per the reviewer's suggestion, we have added more validation of CAR neutrophil off-target toxicity in the discussion.

[redacted to maintain the confidentiality of unpublished data.]

Particularly, off-target toxicity profiling of CAR-neutrophils with or without loading nanodrugs in infused animals, including cytokine release syndrome, neurotoxicity, and on-target off-tumor toxicities observed in CAR-T cells¹⁰, are needed, despite the short lifespan of neutrophils.

3. I would have liked to see overlap between CAR + and NP-loaded neutrophils, since drug loading was apparently around 10% of the neutrophils. What percent were double positive? Do you think this population did the bulk of the work here?

Response: We thank the reviewer for this comment and per the reviewer's suggestion, we have performed flow cytometry analysis of CAR and NP double staining in CAR-neutrophils after loading nanodrugs. Our new data indicates >95% CAR-neutrophils co-

express CAR construct and NP (please see updated **Fig. S7B** and also figure below). We'd also like to apologize for the confusing statement on 10% drug loading efficiency in **Fig. 3D**, and just to clarify, this 10% efficiency refers to the efficiency of loading TPZ into SiO₂ nanoparticles. After centrifuge to remove free TPZ small molecule, the final SiO₂-TPZ nanodrugs were efficiently loaded into CAR-neutrophils (>95%).

[redacted to maintain the confidentiality of unpublished data.]

5. It looks like the bulk RNA sequences methods were left out of the methods section (Figure 4)- I was looking to see if the authors had any information on their bulk RNA seq data. I am curious as to what measures they took in the pipeline to ensure that only tumor cell RNA data was used out of those co-cultures, because I'm wondering if some of their hits are from dying neutrophils instead of the tumor cells.

Response: We thank the reviewer for this comment and apologize for missing the necessary bulk RNA-seq methods in our previous submission. Per the reviewer's suggestion, we have added this part in the Methods section and also copied below. Since neutrophils are floating cells, we washed the cell cultures with PBS buffer five times to

remove residual neutrophils from attached U87MG cells after a short-term 5-hr co-culture¹¹⁻¹⁴. We checked the expression of key neutrophil markers in our RNA-seq data of GBM cells and neutrophils (same batch of RNA-seq data collection), and the low expression of neutrophil markers in GBM bulk RNA-seq data indicates a relatively pure GBM population (please see figure below).

Response Figure 1. Heatmap shows expression levels of neutrophil markers in indicated cells.

Bulk RNA sequencing and data analysis. Bulk RNA sequencing (RNA-seq) was performed according to our previous study [29]. Briefly, neutrophil and U87MG cell cocultures were washed with PBS for five times to remove residual floating neutrophils, and attached U87MG cells were dissociated with 0.25% Trypsin-EDTA solution for total RNA isolation using Direct-zol RNA MiniPrep Plus kit. RNA samples were then prepared and sequenced in Illumina HiSeq 2500 by the Center for Medical Genomics at Indiana University. The resulting sequencing reads were mapped to the human genome (hg 19) using HISAT2 program, and the RefSeq transcript levels (RPKMs) were then quantified using the python script rpkmforgenes.py. Heatmaps of selected gene subsets after normalization were plotted using Morpheus (Broad Institute). The original fastq files and processed RPKM text files were available in NCBI with the GEO accession number GSE206170.

6. There appears to be a labeling error in Figure 7H graph.

Response: We thank the reviewer for this comment and apologize for the mis-labeling in **Fig. 7H**. Per the reviewer's suggestion, we corrected our label in **Fig. 7H** and also copied below:

7. In Figure 5, they show the CAR neutrophils penetrating GBM spheroids. What would have made this figure more convincing is some IHC proof of a hypoxic gradient created within these spheroids to prove activity of the TPZ release and cytotoxicity.

Response: We thank the reviewer for this comment, and per the reviewer's suggestion, we performed immunostaining analysis on HIF-1a expression in GBM spheroids. As compared to normoxic condition, GBM spheroids cultured in hypoxic incubator expressed high levels of HIF-1a (please see **Fig. S12C** and also figure below).

Fig. S12C. Immunostaining analysis of HIF-1 α expression in normoxic and hypoxic GBM spheroids.

8. In Figure S7, how were immunological synapses quantified?

Response: We thank the reviewer for this comment and apologize for missing key information regarding immunological synapse quantification. According to a previous published method¹⁵, we labelled CAR-neutrophils (with or without nanodrug) with anti-CD45-APC antibody and tumor cells with Calcein-AM fluorescent dye before co-culture. Following co-incubation for 0, 2, 4 and 6 hours, cell mixtures were fixed and analyzed in an Accuri C6 plus cytometer (Beckton Dickinson) for double-positive cell analysis. APC+/Calcein+ cells were quantified in FlowJo software as our quantitative immunological synapse formation (now **Fig. S8**). We also added this information to the updated Methods section.

Reviewer #4 (Remarks to the Author):

In this manuscript, the authors developed CAR-neutrophils loaded with therapeutic nanoparticles for glioblastoma therapy. Given the resistance of neutrophils to genomic engineering, the authors decided to create CAR-expressing human pluripotent stem cells and then induced their differentiation into CAR-neutrophils. The authors used these CAR-neutrophils as whole cell carriers to encapsulate silica nanoparticles loaded with hypoxia-responsive chemicals for cancer therapy. Through a series of in vitro studies, they demonstrated that the drug-loaded CAR-neutrophils targeted brain tumor cells, released therapeutic cargo in response to hypoxic conditions, and induced cancer killing. In an in vivo mouse glioblastoma model, drug-

loaded CAR-neutrophils, following systemic administration, were shown to accumulate in the brain, inducing anti-cancer effects and improving the mouse survival rate. Overall, the concept presented in this work is interesting. The authors have put in considerable effort in addressing all prior comments. However, the revised manuscript, as it stands, still contains some issues, as detailed in the comments below.

Response: We would like to thank the reviewer for the nice summary and appreciation of our revised work. We are also grateful for the valuable comments to substantially improve our manuscript, and per the reviewer's suggestion, we have performed additional experiments to address the reviewer's concerns raised from previous submission.

1. Using whole cells as carriers to deliver drug-loaded nanoparticles is not as straightforward as what may have been suggested in this manuscript. The authors did not provide sufficient information on how to load nanoparticles into cells. For example, did the authors try different nanoparticle input to optimize loading yield? Did they optimize the length of time for co-incubating cells and nanoparticles? Also, following the cellular uptake, nanoparticles may undergo degradation inside the cells. Did the authors investigate the kinetics of intracellular degradation of nanoparticles?

Response: We thank the reviewer for this comment and apologize for not clearly presenting our data on nanoparticle characterization. Per the reviewer's suggestion, we performed more characterization in this revision. For loading nanodrugs into neutrophils, CAR-neutrophils (1×10^5 cells/mL) were seeded into a DNA low-bind tube and incubated with nanodrugs (equivalent to 400 $\mu\text{g/mL}$ of SiO_2 content) for 1 hour at 37°C in 5% CO_2

humidified atmosphere. Time-dependent nanodrug loading analysis was also performed (Fig. S7A) and the maximum loading content was reached at 1 hour after cell-nanoparticle incubation. We also performed kinetics of intracellular degradation of nanodrugs (Fig. S9A-C) and showed these nanodrugs were kept stable within neutrophils during neutrophil migration and coculture with tumor cells. After uptaken by tumor cells, nanodrugs gradually degrade within tumor cells (Fig. S6F-G). In this study, we rationally designed and used the GHS-responsive, biodegradable mesoporous organic silica nanoparticles to reduce off-target drug release upon neutrophil apoptosis, and we should have tested and compared the silica nanoparticles with other nanoparticles like cationic liposomes. In future studies, we will optimize the nanoparticles to maximize the drug loading and achieve maximum therapeutic efficacy. We have discussed this limitation in our updated Discussion section and copied below:

Future studies on testing other nanoparticles may yield optimized drug loading in neutrophils and achieve maximum *in vivo* therapeutic efficacy.

2. In the introduction section, the authors made some questionable statements. For example, the authors claimed that “Neutrophils’ innate immunity and plasticity against various pathogens and cancers, including GBM, were not previously explored in drug delivery systems”. This is plainly inaccurate, as many neutrophil-based therapeutic modalities have emerged for treating various diseases including glioblastoma.

Response: We thank the reviewer for this comment and apologize for making questionable statements. Per the reviewer’s suggestion, we have modified our statements and also copied below.

Neutrophils' innate immunity and plasticity against various cancers [12–16], including GBM, were less explored than their application as cell carriers in drug delivery [8–10].

3. In the in vitro tumor penetration study, are the images shown in Figure 5I the cross sections of tumor spheroids? Did the authors quantify the penetration depth of drug-loaded CAR-neutrophils? Also, the authors state that “R-SiO₂-TPZ NPs were encapsulated stably in the CAR-neutrophils during tumor infiltration”. This statement is not convincing, as neutrophils can theoretically interact with tumor cells and undergo cytolysis at any point during tumor penetration.

Response: We thank the reviewer for this comment and apologize for the confusing statement on the nanoparticle stability in neutrophils. Per the reviewer's suggestion, we quantified the penetration depth of drug-loaded CAR-neutrophils within tumor spheroids (please see updated **Fig. S12A-B** and also figure below). In addition, we modified our text to avoid the confusion: R-SiO₂-TPZ NPs were encapsulated stably in the CAR-neutrophils during tumor infiltration **before their cytolysis**.

4. The authors added a new therapeutic efficacy study in Figures 7G-H in the revised

manuscript and noted considerably improved efficacy compared with their previous study in Figure 7. It seems that the only difference between the old and new study is the total number of doses was increased from 4 to 6 in the new study. But given that those extra two doses were added on 32 and 39 days, respectively, why did the survival rates show considerable improvement compared with the old study even before 32 days?

Response: We thank the reviewer for this comment and apologize for the confusing presentation of our data in previous **Fig. 7**. As pointed out by Reviewer #1, the short lifespan of neutrophils significantly affects their *in vivo* antitumor efficacy after intravenous injection. Therefore, increased lifespan of neutrophils in the first 4 doses (before additional doses), as a result of much shorter cell preparation time (in total ~1 hr vs 4 hr) from less experimental groups and less cell isolation during our animal study in **Fig. 7G-H** as compared to **Fig. 7A-D**, increased animal survival in tumor-bearing mice in **Fig. 7G-H**. Furthermore, the additional doses of neutrophils further extended animal survival. While a similar survival curve of the R-SiO₂-TPZ group was observed between these two independent animal studies, we'd also like to acknowledge the potential contribution of batch variability to the different survival rate observed in the CAR-neutrophil experimental groups. We have updated our text to be clearer and also copied below.

While a similar survival curve of the R-SiO₂-TPZ group was observed between these two independent animal studies, reduced time in cell isolation and preparation for injection from a total of ~4 hrs to 1 hr during the first 4 neutrophil doses led to improved animal survival in CAR-

neutrophil groups before day 32. Collectively, our data demonstrated the importance of neutrophil preparation and dosage optimization in future clinical application of neutrophil therapeutics.

Reference:

1. Takano, T., Sada, K. & Yamamura, H. Role of protein-tyrosine kinase Syk in oxidative stress signaling in B cells. *Antioxidants and Redox Signaling* (2002) doi:10.1089/15230860260196335.
2. Zhang, J. *et al.* ROS and ROS-Mediated Cellular Signaling. *Oxidative Medicine and Cellular Longevity* (2016) doi:10.1155/2016/4350965.
3. Kawakami, Y. *et al.* A Ras activation pathway dependent on Syk phosphorylation of protein kinase C. *Proc. Natl. Acad. Sci. U. S. A.* (2003) doi:10.1073/pnas.1633695100.
4. Mócsai, A., Ruland, J. & Tybulewicz, V. L. J. The SYK tyrosine kinase: A crucial player in diverse biological functions. *Nature Reviews Immunology* (2010) doi:10.1038/nri2765.
5. Che, J. *et al.* Neutrophils Enable Local and Non-Invasive Liposome Delivery to Inflamed Skeletal Muscle and Ischemic Heart. *Adv. Mater.* (2020) doi:10.1002/adma.202003598.
6. Xue, J. *et al.* Neutrophil-mediated anticancer drug delivery for suppression of postoperative malignant glioma recurrence. *Nat. Nanotechnol.* (2017) doi:10.1038/nnano.2017.54.
7. Bae, H. B. *et al.* Vitronectin inhibits neutrophil apoptosis through activation of integrin-associated signaling pathways. *Am. J. Respir. Cell Mol. Biol.* (2012) doi:10.1165/rcmb.2011-0187OC.
8. Liu, Y. *et al.* Intracellular Mutual Promotion of Redox Homeostasis Regulation and Iron Metabolism Disruption for Enduring Chemodynamic Therapy. *Adv. Funct. Mater.* (2021) doi:10.1002/adfm.202010390.
9. Liu, B. *et al.* A Tumor-Microenvironment-Responsive Nanocomposite for Hydrogen Sulfide Gas and Trimodal-Enhanced Enzyme Dynamic Therapy. *Adv. Mater.* (2021) doi:10.1002/adma.202101223.
10. Larson, R. C. & Maus, M. V. Recent advances and discoveries in the mechanisms and functions of CAR T cells. *Nat. Rev. Cancer* **21**, 145–161 (2021).
11. Hara, T. *et al.* Interactions between cancer cells and immune cells drive transitions to mesenchymal-like states in glioblastoma. *Cancer Cell* (2021) doi:10.1016/j.ccell.2021.05.002.
12. Jerby-Arnon, L. *et al.* A Cancer Cell Program Promotes T Cell Exclusion and Resistance to Checkpoint Blockade. *Cell* (2018) doi:10.1016/j.cell.2018.09.006.
13. Zhai, Y. *et al.* Single-Cell RNA-Sequencing Shift in the Interaction Pattern Between Glioma Stem Cells and Immune Cells During Tumorigenesis. *Front. Immunol.* (2020) doi:10.3389/fimmu.2020.581209.
14. Finotello, F. *et al.* Molecular and pharmacological modulators of the tumor immune contexture revealed by deconvolution of RNA-seq data. *Genome Med.* (2019) doi:10.1186/s13073-019-0638-6.
15. Matlung, H. L. *et al.* Neutrophils Kill Antibody-Opsonized Cancer Cells by Trogoptosis. *Cell Rep.* (2018) doi:10.1016/j.celrep.2018.05.082.
16. Fridlender, Z. *et al.* Polarization of tumor-associated neutrophil phenotype by TGF-beta:

- "N1" versus "N2" TAN. *Cancer Cell* (2009).
17. Blaisdell, A. *et al.* Neutrophils Oppose Uterine Epithelial Carcinogenesis via Debridement of Hypoxic Tumor Cells. *Cancer Cell* (2015) doi:10.1016/j.ccell.2015.11.005.
 18. Mahiddine, K. *et al.* Relief of tumor hypoxia unleashes the tumoricidal potential of neutrophils. *J. Clin. Invest.* (2020) doi:10.1172/JCI130952.
 19. Yan, J. *et al.* Human polymorphonuclear neutrophils specifically recognize and kill cancerous cells. *Oncoimmunology* **3**, e950163 (2014).
 20. Lin, Y. J., Wei, K. C., Chen, P. Y., Lim, M. & Hwang, T. L. Roles of Neutrophils in Glioma and Brain Metastases. *Frontiers in Immunology* (2021) doi:10.3389/fimmu.2021.701383.
 21. Wu, M. *et al.* MR imaging tracking of inflammation-activatable engineered neutrophils for targeted therapy of surgically treated glioma. *Nat. Commun.* **9**, 1–13 (2018).
 22. Chu, D., Dong, X., Zhao, Q., Gu, J. & Wang, Z. Photosensitization Priming of Tumor Microenvironments Improves Delivery of Nanotherapeutics via Neutrophil Infiltration. *Adv. Mater.* **29**, (2017).

REVIEWERS' COMMENTS

Reviewer #1 (Remarks to the Author):

The authors adequately addressed all of my concerns.

Reviewer #3 (Remarks to the Author):

The authors have satisfactorily addressed my concerns

Reviewer #4 (Remarks to the Author):

The authors have adequately addressed my comments.

Reviewer #1 (Remarks to the Author):

The authors adequately addressed all of my concerns.

Response: We would like to thank the reviewer for the appreciation of our revised work and we're glad to hear that the reviewer's concerns are fully addressed.

Reviewer #3 (Remarks to the Author):

The authors have satisfactorily addressed my concerns.

Response: We would like to thank the reviewer for the appreciation of our revised work and we're glad to hear that the reviewer's concerns are fully addressed.

Reviewer #4 (Remarks to the Author):

The authors have adequately addressed my comments.

Response: We would like to thank the reviewer for the appreciation of our revised work and we're glad to hear that the reviewer's concerns are fully addressed.